

# A process-based model for fluvial valley width

Jens M. Turowski[1], Aaron Bufe[1,2], Stefanie Tofelde[3]

[1] Helmholtz Zentrum Potsdam, GeoForschungsZentrum (GFZ) Potsdam, Potsdam, Germany

[2] Department of Earth and Environmental Sciences, Ludwig Maximilian University Munich, Munich, Germany

[3] Institute of Geosciences, University of Potsdam, Potsdam, Germany

*Correspondence to*: Jens M. Turowski (jens.turowski@gfz-potsdam.de)

**Abstract.** The width of fluvial valley-floors is a key parameter to quantifying the morphology of mountain regions. Valley-floor width is relevant to diverse fields including sedimentology, fluvial geomorphology, and archaeology. The width of

valleys has been argued to depend on climatic and tectonic conditions, on the hydraulics and hydrology of the river channel that forms the valley, and on sediment supply from valley walls. Here, we derive a physically-based model that can be used to predict valley width and test it against three different datasets. The model applies to valleys that are carved by a river migrating laterally across the valley floor. We conceptualize river migration as a Poisson process, in which the river changes its direction stochastically, at a mean rate determined by hydraulic boundary conditions. This approach yields a characteristic timescale for

the river to once cross the valley floor from one wall to the other. The valley width can then be determined by integrating the speed of migration over this timescale. For a laterally unconfined river that is not uplifting, the model predicts that the channel-belt width scales with river-flow depth. Channel-belt width corresponds to the maximum width of a fluvial valley. We expand the model to include the effects of uplift and lateral sediment supply from valley walls. Both of these effects lead to a decrease in valley width in comparison to the maximum width. We identify a dimensionless number, termed the mobility-uplift number,

which is the ratio between the lateral mobility of the river channel and uplift rate. The model predicts two limits: At high values of the mobility-uplift number, the valley evolves to the channel-belt width, whereas it corresponds to the channel width at low values. Between these limits, valley width is linked to the mobility-uplift number by a logarithmic function. As a consequence of the model, valley width increases with increasing drainage area, with a scaling exponent that typically has a value between 0.4 and 0.5, but can also be lower or higher. We compare the model to three independent data sets of valleys in experimental

and natural uplifting landscapes and show that it closely predicts the first-order relationship between valley width and the mobility-uplift number.

**Plain language summary.** Fluvial valleys are ubiquitous landforms, and understanding their formation and evolution affects a wide range of disciplines, from archaeology over geology to fish biology. Here, we develop a model to predict the width of

fluvial valleys for a wide range of geographic conditions. In the model, fluvial valley width is controlled by the two competing factors of lateral channel mobility and uplift. The model complies with available data and yields a broad range of quantitative predictions.





## 1    Introduction

Many ancient civilizations developed in river valleys (Macklin et al., 2015). There, fertile soil was readily available, and the river provided water, fish, and a transport route. It is a common observation that large rivers often feature broad valley floors with valley floodplains that are several times wider than the river itself (Fig. 1a). Valley-floor width (valley width hereafter) is the width of the valley from foot to foot of the enclosing valley walls, and hence the sum of river width and floodplain width. In fluvial valleys, valley width usually corresponds to the part of the valley in which the river is active on timescales

encompassing multiple floods, and, thus, is intimately related to the width of the channel belt in an unconfined setting without valley walls (Fig. 1b) (e.g., Limaye, 2020; Tofelde et al., 2022). The nearly planar valley floors do not only provide space for settlements and farming grounds, but also accommodate alluvial sediments supplied from upstream mountain regions, and often host unique ecological communities. As such, valley width has been recognized as an important parameter in the development of human settlements (e.g., Hillier et al., 2007; Macklin et al., 2015; Rigsby et al., 2003), the evolution of orogenic

landscapes (e.g., Hancock & Anderson, 2002; Langston & Tucker, 2018), the distribution of sediments in the landscape (e.g., Blöthe et al., 2014; Blum & Törnqvist, 2000; Jonell et al., 2018), the development of river patterns (e.g., Fotherby, 2009; Schumm & Lichty, 1963), floodplain ecology (e.g., Naiman et al., 2010), speciation and biodiversity (e.g., Perrigo et al., 2020; Steinbauer et al., 2016), and the establishment of fisheries (e.g., May et al., 2013).

Multiple parameters have been suggested to control valley width. It has been observed that valley width is correlated to water

discharge, stream length, or drainage area, as well as upstream sediment supply in natural river valleys (e.g., Constantine et al., 2014; Dunne et al., 2010; Salisbury, 1980; Salisbury et al., 1968; Tomkin et al., 2003; Zavala et al., 2021) and analogue experiments (e.g., Bufe et al., 2016a; Martin et al., 2011). Valley width typically scales with discharge or drainage area according to a power law, with scaling exponents that vary between about 0.1 and 1.2 (e.g., Beeson et al., 2018; Brocard & van der Beek, 2006; Langston & Temme, 2019; Snyder et al., 2003; Som et al., 2009; Tomkin et al., 2003). It has also been

observed that valley width is inversely correlated to uplift rate (e.g., Bufe et al., 2016a, Clubb et al., 2023a), and, in the special case of paired alluvial river terrace sequences, inversely correlated to valley-wall height (Tofelde et al., 2022). In addition, for comparable discharge conditions, valleys seem sometimes to be wider in softer lithologies compared to harder lithologies (e.g., Brocard & van der Beek, 2006; Bursztyn et al., 2015; Keen-Zebert et al., 2017; Langston & Temme, 2019; Moore, 1926; Schanz & Montgomery, 2016), and widening rates have been suggested to depend on rock type (e.g., Johnson & Finnegan,

2015; Limaye and Lamb, 2014; Marcotte et al., 2021; Montgomery, 2014; Snyder & Kammer, 2008; Suzuki, 1982). In contrast, in a regional study of the Himalaya, Clubb et al. (2023a) reported that valley width is independent of lithology, and concluded that uplift provides the dominant control.

Multiple authors have suggested that valley widening occurs during times when the river aggrades or moves laterally through a thick sediment fill without major incision (e.g., Maddy et al., 2001; Hancock & Anderson, 2002). Further, it has been argued

that river valleys widen by lateral erosion of streams and by fluvial undercutting of valley-wall hillslopes and their subsequent collapse (Brocard & van der Beek, 2006; Hancock & Anderson, 2002; Martin et al., 2011; Malatesta et al., 2017; Suzuki, 1982). In this case, widening rates decrease with increasing valley width, because the river spends a decreasing fraction of



time in contact with the valley walls (Hancock & Anderson, 2002; Tofelde et al., 2022). However, a steady state is never reached and the valley widens indefinitely. As a result, valley width would be determined by the time since the onset of lateral

migration and erosion, and the widening rate. Some river valleys show paired terrace sequences, which are often attributed to cyclic climate change (e.g., Bridgland & Westaway, 2008; Maddy et al., 2001; Schanz et al., 2018). Their presence implies that valleys can evolve to different widths under similar climatic conditions. To explain the occurrence of paired terraces, Tofelde et al. (2022) argued that a parameter independent of river dynamics is also important in controlling the width to which valleys evolve. They suggested that a steady-state valley width is achieved when lateral sediment supply from hillslopes is

balanced with the ability of the river to remove this sediment. Their model can explain the existence of paired terrace sequences and predicts the observed inverse-linear scaling between width and total height of enclosing valley walls. Yet, the model does not predict how valley width is modulated by uplift, and it can only predict valley width in relation to a maximum valley width that is an input parameter in the equations. Tofelde et al. (2022) suggested that this maximum valley width corresponds to channel-belt width in an unconfined setting. Limaye (2020) postulated that channel-belt width scales with the channel width

of the forming river, which still lacks a mechanistic explanation.

It seems clear that hydraulics and river processes (e.g., Martin et al., 2011; Suzuki, 1982) as well as tectonics (e.g., Bufe et al., 2016a; Clubb et al., 2023a) influence the width of fluvial valleys, while the role of lithology is unclear (cf. Clubb et al., 2023a; Langston & Temme, 2019). Yet, a full understanding of the controls and a model that allows predicting valley width from known boundary conditions is currently missing. In particular, it is not understood how the observed scaling relationships

between valley width, drainage area, and uplift rate arise (e.g., Beeson et al., 2003; Bufe et al., 2016a; Clubb et al., 2023a; Langston & Temme, 2019). Here, we build on previous work of Bufe et al. (2019) and Tofelde et al. (2022), and develop a process-based model for the steady-state width of channel belts and fluvial valleys. The model predicts the width of channel belts in laterally unconfined settings, and how this width is reduced in laterally confined valleys and in uplifting regions. We compare the model to three complementary datasets, of rivers crossing uplifting folds in an experiment (Bufe et al., 2016a)

and the Tian Shan mountain range, and to a valley-width compilation with more than 1.6 million datapoints from the Himalaya (Clubb et al., 2023a,b).

## 2  Model development
### 2.1  Conceptual framework of model

We start by considering the width $W$ [L] of a valley containing an alluvial river (Fig. 1c). We will proceed with the derivation

and make a connection to bedrock river valleys in the discussion. We postulate that the walls of fluvial valleys are pushed back by fluvial undercutting that drives wall collapse, and subsequent evacuation of the resulting sediment when the river is located at the valley wall and moves into it (cf. Hancock & Anderson, 2002; Martin et al., 2011; Malatesta et al., 2017). We assume that processes acting in the long-channel direction are negligible to first order, and that each point of the river can be treated as independent of events upstream and downstream. Thus, we consider a valley cross-section, in which a stream migrates forth

and back across the valley floor with lateral speed $V$ [L T$^{-1}$] (Fig. 1). For a given set of climatic, tectonic and sedimentological boundary conditions, we conceptualize the lateral motion of the channel as a stochastic process, in which switches in the



direction of motion are considered as identically distributed and independent stochastic events occurring at a constant rate. They can therefore be described by a Poisson process with rate parameter $\lambda$ [T$^{-1}$] that quantifies the mean number of switch events per unit time. At the valley walls, the need to erode and transport sediments supplied from valley-wall hillslopes may

slow down the lateral speed of the channel to a value $v < V$ (Tofelde et al., 2022). The valley width is then determined by (i) the speed of lateral migration of the river across the floodplain, (ii) the length of time the river moves on average in the same direction, and (iii) the amount of laterally supplied sediment from hillslopes (cf. Tofelde et al., 2022). For negligible lateral sediment supply, the width $W$ of the valley can be obtained by integrating over the product of the lateral speed of motion $V$ and a characteristic timescale $\Delta t$ [T].

$$W = \int_0^{\Delta t} V dt + W_C.$$

(1)

The width of the river, $W_C$ [L], needs to be added, as it presents the starting condition before any bevelling takes place. Thus, channel width $W_C$ provides a minimum width for the valley. The timescale $\Delta t$ is related to the mean waiting time between switch events. In a Poisson process with rate constant $\lambda$, the waiting times are exponentially distributed with a mean of $1/\lambda$.

Because the process is stochastic, there is a non-negligible probability that the waiting time is larger than the average. As such, the effective lateral migration time that sets valley width can be expected to be slightly larger than the mean waiting time. Therefore, $\Delta t$ is inversely related to $\lambda$ by

$$\Delta t = \frac{c}{\lambda},$$

(2)

where $c$ [-] is a dimensionless constant of order one. We proceed by considering the average behaviour of the channel belt. Thus, the equations yield a well-defined steady state and spatially stable channel belt. In a fully stochastic model, the channel belt would drift laterally once it reaches the steady state width.





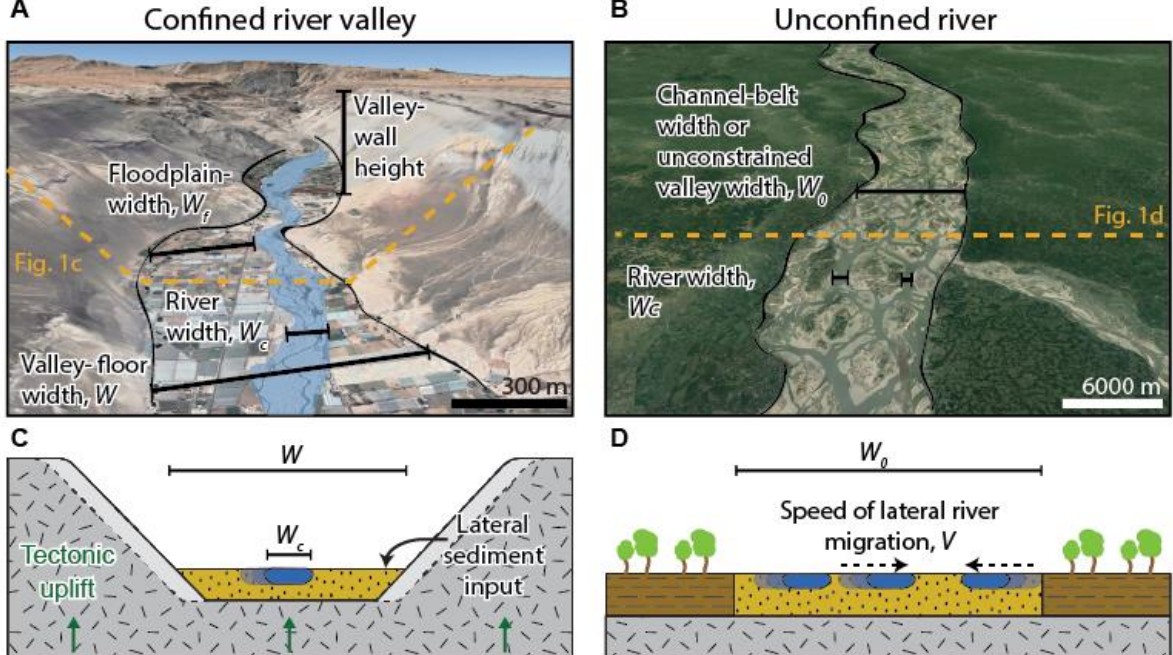

**Fig. 1: Examples and concepts for confined river valleys and unconfined rivers. (A) Oblique view from © Google Earth of the San Jose River, Chile (18.58°S, 69.97°W), showing a confined valley. Debris cones on the valley flanks are signs for substantial sediment input from valley walls into the river valley. The scalebars refer to the foreground. (B) Oblique view from © Google Earth of the Brahmaputra river, Bangladesh (25.3°N, 89.7°E), that is laterally unconfined. (C) Conceptual sketch of the dynamics in a confined river valley. (D) Conceptual sketch for the dynamics in an unconfined channel belt.**


## 2.2    Model derivation

### 2.2.1    Unconfined river: Channel-belt width

To complete the model, we need to provide equations for the channel's lateral speed of migration $V$ and the rate parameter $\lambda$, which we will treat in turn. For the former, we use the concept of Bufe et al. (2019) that states that, for a given discharge,

sediment supply and grain size, the amount of sediment that the channel can move by lateral erosion per unit channel length per unit time is constant and can be expressed by a lateral-transport capacity $q_L$ [$L^2\,T^{-1}$] (Fig. 2). The lateral migration speed, $V$, is then equal to the ratio of $q_L$ and the height of the river bank in the direction of motion, $H_+$ [L] (Bufe et al., 2019):

$$V = \frac{q_L}{H_+.}$$

(3)

For constant boundary conditions without uplift, $H_+$ can be considered as a constant $H_0$ [L], which should be equal to flow depth $h$ [L], because during migration, the channel cannot deposit sediment at elevations higher than its flow depth. Then, eq. (1) can be solved and the width of the channel belt in an unconfined plain, $W_0$, is given by

$$W_0 = \int_0^{\Delta t} V dt + W_C = V\Delta t + W_C = \frac{q_L}{H_0}\Delta t + W_C = \frac{c}{\lambda}\frac{q_L}{h} + W_C.$$





(4)

To quantify the rate parameter $\lambda$, we postulate that the channel switches direction when its cross section is overwhelmed by sediment derived from erosion of the bank in the migration direction, leading to water overflow of the bank opposite of the motion direction (Fig. 2). Hence, the likelihood of channel switching, $\lambda$, is proportional to the ratio of the average sediment input rate due to lateral migration, $q_L$ (yellow shaded area in Fig. 2), and the dimensions of the channel given by the product of channel width and flow depth, $W_c h$ (blue shaded area in Fig. 2). Thus, we suggest that $\lambda$ scales as:

$$\lambda \propto \frac{q_L}{W_c h}.$$

(5)

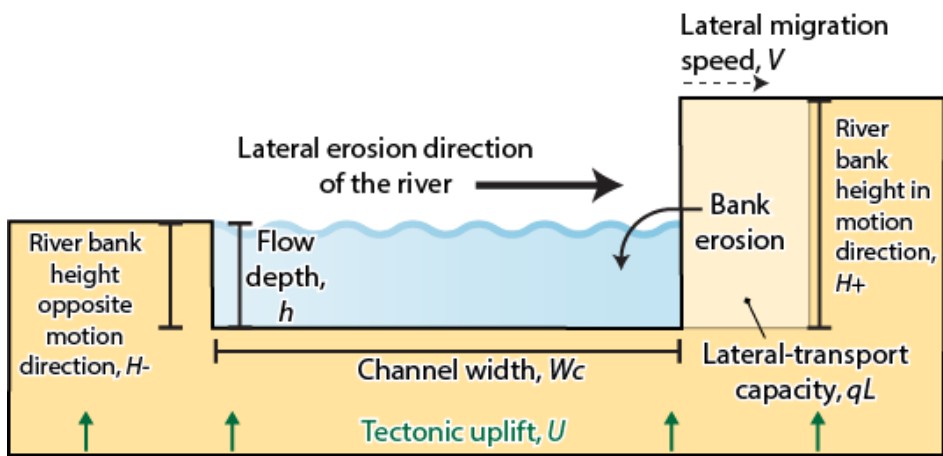

Fig. 2: Schematic cross section of a migrating channel, with definition of parameters.


However, eq. (5) is not a complete description of the scaling. We expect that $\lambda$ depends not only on the sediment input rate relative to the channel cross-sectional area, but also on the aspect ratio of the channel. In particular, we suggest that deep and narrow channels are less likely to switch directions than wide and shallow channels for otherwise similar conditions. Wide and shallow channels have a lower bank relative to the channel dimensions and lateral-transport capacity, which should make

switching directions more likely. Therefore, we expect that lambda scales with the aspect ratio as

$$\lambda \propto \frac{W_c}{h}.$$

(6)

Combining eqs. (5) and (6) gives the final relation for the rate parameter $\lambda$:

$$\lambda = k \frac{q_L}{h^2}.$$

(7)

Here, $k$ [-] is a dimensionless constant. Substituting eq. (7) into eq. (4) yields the channel-belt width in unconfined settings



$$W_0 = \frac{k_0}{H_0}h^2 + W_C = k_0 h + W_C.$$

(8)

Here, $k_0 = c/k$ [-] is a dimensionless constant, and we assumed $H_0 = h$ in the latter identity. The channel-belt width $W_0$ predicted by eq. (8) at the same time gives the maximum valley width in the absence of uplift and lateral hillslope sediment supply.

### 2.2.2    Confined valleys in uplifted regions

Here, we consider a river incising at a constant rate. The incision may be driven by relative uplift, a change in water and sediment discharge, or autogenic variations in river dynamics. We proceed with the derivation considering the case of uniform uplift, noting that the results should be equivalent for any other process driving river incision.

In an uplifted region, the river adjusts to a state in which the incision rate equals the uplift rate (e.g., Howard, 1994; Turowski, 2020). Yet, the parts of the valley floor where the channel is not currently located rise in elevation at the uplift rate $U$ [L T$^{-1}$]. As the river migrates laterally, it therefore needs to remove the additional sediment material provided due to uplift. The amount of this sediment at a particular location scales with the product of the uplift rate and the time since the last visit of the river at that location. We can model this as an increase in the bank height $H_+$ encountered by the river, which is given by

$$\frac{dH_+}{dt} = U.$$

(9)

Within the integral (eq. 1), we thus need to treat $H_+$ as a time-dependent parameter. The integral can be solved by a substitution of variables to yield valley width $W$:

$$W = \int_0^{\Delta t} \frac{q_L}{H_{+(t)}}\, dt + W_C = \int_{H_0}^{H_0 + 2U\Delta t} \frac{q_L}{UH_+}\, dH_+ + W_C = \frac{q_L}{U}\ln\left\{1 + \frac{2U\Delta t}{H_0}\right\} + W_C$$

(10)

Here, $\ln\{x\}$ denotes the natural logarithm of $x$. The factor of two in the upper limit of the integral arises because the river needs to switch direction and traverse the valley twice before arriving at the same position again. Therefore, the time elapsed between revisiting a valley margin is $2\Delta t$. Assuming that the river-cross-sectional shape is unaffected by uplift, the timescale $\Delta t$ is the same as in the unconfined case (cf. eqs. 2&7). As noted above, for consistency, we need to substitute $2\Delta t$. Then, using eqs. (7) and (8), $W$ is given by

$$W = \frac{q_L}{U}\ln\left\{1 + \frac{U(W_0 - W_C)}{q_L}\right\} + W_C.$$

(11)

For $U = 0$, eq. (11) reduces to $W = W_0$, as required, and for large $U$, $W = W_C$, as can be expected.



### 2.2.3 Sediment supply from valley walls

To explain the geometry of paired river terraces, Tofelde et al. (2022) suggested that lateral sediment supply from hillslope erosion or back-weathering processes leads to valley narrowing, because a river can only widen the valley further once this additional material deposited in sediment cones at the wall toe is evacuated. Tofelde et al. (2022) proposed that valleys reach

a steady state width, at which lateral sediment removal by the river equals lateral sediment input from hillslopes. This lateral mass balance can be written as

$$Pq_L = q_H.$$

(12)

Here, $P$ [-] is the fraction of time that the river spends cutting into the channel walls, and $q_H$ [L$^2$ T$^{-1}$] is the rate of sediment

supply from the valley walls per unit channel length. In their proposed valley-width model, Tofelde et al. (2022) derived $P$ under the assumption that the channel width is much smaller than the valley width and can therefore be neglected. Including channel width in the derivation of $P$ yields (compare to eqs. 10 to 14 of Tofelde et al., 2022)

$$P = \frac{W_0 - W}{W_0 - W_C}.$$

(13)

After substituting eq. (13) into eq. (12) and solving for $W$, valley width is given by

$$W = W_0 - \frac{q_H}{q_L}(W_0 - W_C).$$

(14)

Note that this equation is defined only as long as $q_H < q_L$. If lateral sediment supply exceeds the capacity of the river to transport the sediment, the river will either aggrade and steepen to increase $q_L$, or will change course and abandon the valley

(Humphrey & Konrad, 2000). Equation (14) updates the model of Tofelde et al. (2022) to include a finite channel width, but excludes uplift. In an uplifting region, $W_0$ in eq. (14) can be identified with $W_U$, and after substituting eq. (11), we obtain an equation for valley width including both uplift and lateral sediment supply:

$$W = \left(\frac{q_L - q_H}{U}\right)\ln\left\{1 + \frac{U(W_0 - W_C)}{q_L}\right\} + W_C.$$

(15)

For $U = 0$, eq. (15) reduces to eq. (14), and for large $U$, $W_0 = W_C$.

We can formulate a non-dimensional version of eq. (15), including four non-dimensional parameters: a valley width normalized to the unconfined channel-belt width $W' = W/W_0$, a channel width normalized to the unconfined channel-belt width $W_C' = W_C/W_0$, a hillslope sediment supply normalized by the lateral-transport capacity $q_H' = q_H/q_L$, and a mobility-uplift number that describes the lateral transport capacity of the river relative to the uplift flux across the valley $M_U = q_L/UW_0$:


$$W' = \frac{W}{W_0} = (1 - q_H')\ln\left\{1 + \frac{1 - W_C'}{M_U}\right\}M_U + W_C'.$$

(16)





Our model provides the first process-based analytical model for channel-belt width in unconfined settings (eq. 8), and valley widths impacted by rock uplift and subject to lateral sediment input from hillslope processes (eqs. 15 and 16).

## 3     Model tests

We test the model predictions with three data sets of valleys forming in uplifting landscapes across different scales and under different boundary conditions (Fig. 3). First, we use existing experimental data of valleys across a single uplifting fold (Bufe et al., 2016a) (Fig. 3a). These experiments isolate the role of uplift on valley formation under controlled boundary conditions. Second, we collected a dataset of valleys formed across uplifting folds in the foreland of the Tian Shan (NW China) to complement the experimental dataset (Fig. 3b,c). Third, we use a recent compilation of more than 1.6 million valley widths

from the Himalaya (Clubb et al., 2023b) (Fig. 3d). None of the datasets contain direct measurements for all model parameters, and each dataset needs a unique approach to defining the necessary proxies. To test our new model, we start with eq. (15) and write it as

$$W = am\ln\left\{1 + \frac{\overline{W_0} - \overline{W_C}}{am}\right\} + \overline{W_C}.$$

(17)

Here, $m$ is a proxy that scales with the ratio of lateral transport-capacity to uplift $q_L/U$ and that can differ between the data sets. The factor $a$ is a scaling parameter linking the proxy data to that ratio. $\overline{W_0}$ and $\overline{W_C}$ are the average unconfined channel-belt width and the channel width, respectively. In each model test, we treat $m$ as the independent variable, $W$ as the dependent variable and $a$, $\overline{W_0}$ and $\overline{W_C}$ as free fit parameters. For individual data points within one data set, $W_0$ and $W_C$ likely vary. However, in the limits of low and high uplift rate, the model equation (eq. 15) converges to $W_0$ and $W_C$, respectively. These

limits insure that the effective fitted values for $\overline{W_0}$ and $\overline{W_C}$ converge to the true means of valley and channel width, respectively. Note that we do not treat the hillslope sediment supply $q_H$ as a separate fit parameter, because it would largely affect the effective value of $\overline{W_0}$ (compare to eq. 16).

### 3.1     Test 1: Experiments on channels crossing a fold

One of the simplest systems to isolate the control of uplift on valley width is to study the narrowing of valleys across single well-defined zones of uplift. Bufe et al. (2016a) conducted six experiments of braided alluvial channels crossing a single uplifting fold (Fig. 3a). These experiments were conducted in a stream-basin with dimensions of 4.8×3.0×0.6 m (Fig. 3a). The basin was filled with well-sorted silica sand ($D_{50} = 0.52$ mm). A flexing metal sheet underneath the basin allowed the uplift of a ~0.5 m-wide zone across the entire basin, forcing the river to cross the uplifting zone. At the start of the experiments, the

river system built a braided channel network and aggraded rapidly. Once the average rate of aggradation across the basin dropped to <10-20% of the input sediment discharge, the fold was uplifted in increments of ~4 mm. Across the six experiments, uplift rates varied by two orders of magnitude. In turn, water and sediment discharges were kept constant with the exception of one experiment with lower sediment discharge (Table 1).



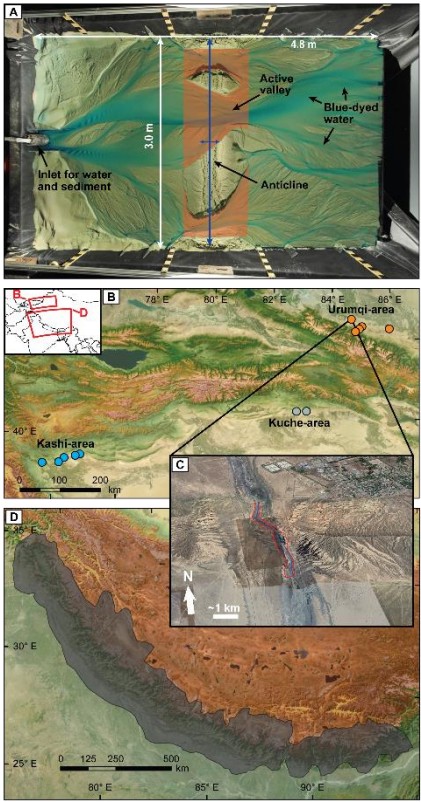

**Fig. 3: Overview of the datasets used for model testing. (A) Overhead picture from an analogue experiment of braided rivers that cross an active uplift. The red shaded area marks the area of the uplift eroded by streams that is divided by the length of the area in the downstream direction to obtain a characteristic width. Where the stream splits, the entire valley area is summed. Figure adapted from Bufe et al. (2016a). (B) Locations of folds in the foreland of the Tian Shan for which we assembled uplift rates from the literature and mapped valley widths on Google Earth. Basemap sourced from Esri and hillshade created from an SRTM digital elevation model (C) Oblique © Google Earth View of the Dushanzi anticline (location in B) and the mapped area (red) and stream length (blue) across the valley. (D) Overview of the Himalaya and the area covered by the width dataset of Clubb et al. (2023b). Basemap sourced from Esri and hillshade created from an SRTM digital elevation model.**

Testing the model provided in eq. (15) requires the quantification of valley width, $W$, and the ratio of $q_L/U$ or its proxy $m$ from the experimental data. Mean valley width was calculated from the bevelled area of the fold divided by the length of the uplifted area of 0.5 m (Table 1). As $U$ was set as a boundary condition for each run, only the lateral-transport capacity, $q_L$, needs to be estimated from other measured experimental parameters. Because many channel parameters, such as channel width and depth, are ill-defined in the quickly evolving braided river system of the experiment, we need to define effective parameters such as representative means for a comparison with eq. (15). Bufe et al. (2016a) measured the area that was actively reworked by channels prior to uplift, $A_f$ [L²], and they determined a timescale over which this active area was reworked, $T_f$ [T]. As such, $A_f$ is equivalent to the area covered by the channel belt (Fig. 1b). Bufe et al. (2019) measured the bank height prior to uplift, $H_0$, on the scale of the experiment. The volume of sediment that is reworked laterally then scales with the ratio of the actively reworked area times the channel-bank height in the part of the experiment without uplift, and the channel mobility timescale





$(A_f H_0/T_f)$. When normalized by the length of the channel system for which these parameters are constrained, $L = 0.88$ m, we

obtain a proxy for the amount of sediment that the channel can move laterally per unit channel length per unit time, $q_L$. Finally, we need to divide by uplift rate $U$ to obtain $m$ as

$$\frac{q_L}{U} \propto m = \frac{A_f H_0}{L T_f U}.$$

(18).

In all experiments, the average valley width across the fold was estimated as the total eroded area (red-shaded area in

Fig. 3a) divided by the length of the fold in the downstream direction (see Bufe et al., 2016a for details), which is equivalent to summing the width across all individual valleys.

**Table 1: Experimental data used for Test 1 (Bufe et al. 2016).**

| Run | Uplift rate $U$ / $10^{-6}$m/s | Sediment supply / ml/h | Water input / ml/h | $T_f$ / h | $A_f$ / m$^2$ | $H_0$ / mm | $m$ / $10^3$ m | $W$ / m |
|-----|-----|-----|-----|-----|-----|-----|-----|-----|
| 1 | 40.0 | 15.8 | 790 | 0.2 | 2.3 | 4.3 | 1.40 | 1.22 |
| 2 | 8.00 | 15.8 | 790 | 0.5 | 2.3 | 4.8 | 2.75 | 1.73 |
| 3 | 4.00 | 15.8 | 790 | 0.3 | 2.3 | 4.5 | 8.10 | 1.92 |
| 4 | 0.40 | 15.8 | 790 | 0.4 | 2.1 | 4.5 | 63.9 | 2.59 |
| 5 | 4.00 | 2.4 | 790 | 0.8 | 1.8 | 7.9 | 3.92 | 1.51 |
| 6 | 4.00 | 15.8 | 790 | 1.3 | 0.5 | 10.0 | 1.53 | 0.92 |


### 3.2    Test 2: Channels crossing folds in the Tian Shan foreland

To complement the experimental dataset, we extracted widths of valleys crossing single uplifting folds in the desert foreland of the Tian Shan, NW China (Fig. 3b-c). The Tian Shan is a major intracontinental mountain range that features uplift rates of ~20-25 mm/yr and accommodates an equivalent of 40-60% of the total convergence between the Indian and Eurasian plates

(Abdakhmatov et al., 1996; Zubovich et al., 2010, 2016). Along the southern and northern foreland, a series of detachment-, fault-bend and fault-propagation folds have uplifted the Cenozoic clastic basin fill of the Tarim and Junggar basins and are incised by antecedent streams that drain the Tian Shan (Avouac et al., 1993; Bufe et al., 2017a, 2017b, Chen et al., 2007; Heermance et al., 2007, 2008; Hubert-Ferrari et al., 2007; Li et al., 2012, 2013, 2015, Scharer et al., 2004; Tapponier & Molnar, 1979; Thompson Jobe et al., 2017). Along both the southern and northern Tian Shan, we selected 12 channels crossing active

folds for which kiloyear uplift rates have been estimated by a combination of optically-stimulated luminescence dating and cosmogenic nuclide dating of deformed terraces (Table 2) (Bufe et al., 2017b; Gong et al., 2014; Li et al., 2015; Lu et al.,





2017; Malatesta et al., 2018; Thompson, 2013). Rivers in the Kashi area (Fig. 3b), the Kezile River on the eastern Quilitage, and the Kuitun River crossing the Dushanzi anticline (Table 1), incise weakly-consolidated late Miocene to Pleistocene sand-, silt-, and mudstones (Chen et al., 2007; Heermance et al., 2007, 2008; Scharer et al., 2004). In turn, the other valleys include

deeper, older, and more indurated clastic sediments that include conglomerates. Precipitation rates are poorly constrained in the Tian Shan, but folds in the Kuche and Urumqi areas are crossed by streams that generally receive more precipitation than streams north of Kachi (Fan et al., 2020). Across all structures, we mapped the valley floor and centerlines by hand on Google Earth imagery and obtained an estimate of a characteristic valley width from the ratio of valley-floor area to the length of the valley center line. In the case of the Boguzihe River crossing the central Atushi fold, the width of the valleys across both

tributaries that cross the fold were summed. This measurement is equivalent to the method of estimating valley width from the experimental data (Bufe et al., 2016a).

To compare these measurements with the model equation, we assumed that the lateral transport capacity per unit channel length scales with drainage area, $A$ [$L^2$]. This assumption is consistent with experimental observations of a near linear scaling of lateral transport capacity and water discharge (Bufe et al., 2019; Wickert et al., 2013). As such, the proxy parameter can be

calculated as

$$\frac{q_L}{U} \sim m = \frac{A}{U}.$$

(19)



**Table 2: Data for the Tian Shan channels, used for Test 2**

| River | Area | Group | Fold | Latitude | Longitude | Drainage area / km² | Uplift rate / mm/yr | Mean valley width / m | Reference |
|---|---|---|---|---|---|---|---|---|---|
| Unnamed | Kachi | South | Mutule | 39.910742 | 76.547338 | 60 | 1.9±0.5 | 75.7 | Bufe et al. (2017b) |
| Boguzihe | Kachi | South | Atushi Central | 39.725553 | 76.119586 | 4280 | 1.0±0.3 | 663.7 | Bufe et al. (2017b) |
| Baishikeremu he | Kachi | South | Kashi | 39.593351 | 75.969232 | 3400 | 2.7±0.7 | 493.8 | Bufe et al. (2017b) |
| Kalanggouluk ehe | Kachi | South | Mingyaole | 39.511221 | 75.443420 | 1910 | 2.7±1.6 | 276.6 | Li et al. (2015), Thompson (2013) |
| Unnamed | Kachi | South | Atushi East | 39.845321 | 76.451615 | 80 | 1.0±0.3* | 78.7 | Bufe et al. (2017b) |
| Bositankelake | Kuche | North | East Qiulitage | 41.87154 | 83.33694 | 654 | 0.80±0.04 | 414 | Zhang et al., 2021 |
| Kezile | Kuche | North | East Qiulitage | 41.90745 | 83.66205 | 321 | 1.6±0.3 | 334 | Zhang et al., 2021 |
| Manas | Urumqi | North | Mana | 44.18788 | 86.12354 | 5541 | 13.5±0.6 | 411 | Gong et al., 2014 |
| Kuitun | Urumqi | North | Dushanzi | 44.32030 | 84.78589 | 2016 | 10.7±1.3 | 333 | Malatesta et al., 2018 |
| Anjihai | Urumqi | North | Nananjihai | 44.10282 | 85.10027 | 1173 | 47±16 | 253 | Malatesta et al., 2018 |
| Anjihai | Urumqi | North | Huoerguos & Nananjihai | 44.16894 | 85.17422 | 1466 | 47±56 | 335 | Lu et al., 2017 |
| Anjihai | Urumqi | North | Nananjihai south | 44.02634 | 84.97666 | 1063 | 44.4±0.6 | 240 | Lu et al., 2017 |

* In the absence of kiloyear uplift rates, these rates are assumed to be similar to Atushi Central

### 3.3 Test 3: Valley width data set from the Himalaya

Clubb et al. (2023a,b) measured valley width at more than 1.6 million locations in the Himalaya (Fig. 3d) using the method of
Clubb et al. (2022), together with some auxiliary data that can be derived from topography. These include drainage area $A$,
channel bed slope $S$, and the normalized steepness index $k_{sn}$, which is a measure of the slope of the channel normalized by the
drainage area (e.g., Kirby & Whipple, 2012; Wobus et al., 2006). Clubb et al. (2023a,b) used SRTM data with a pixel size of
30 m, and can therefore only measure valley width with a minimum width of approximately two pixels or 60 m.

Similar to the Tian Shan data (section 3.2), we assume that $q_L$ scales with drainage area. Uplift rates are not available. However,
it has been shown that the normalized steepness index broadly scales with measured erosion rates in the Himalaya and in other
mountain ranges (e.g., Kirby & Whipple, 2001; Lague, 2014; Wobus et al., 2006). In turn, erosion rates are a first order proxy
for uplift in the Himalaya. Here, we assume that the relationship between uplift rate and normalized steepness index $k_{sn}$ [L$^{0.9}$]
is linear. Even though relationships between $k_{sn}$ and erosion rate are commonly fit with non-linear power laws, the scatter in
most data sets make a linear fit equally appropriate (Kirby & Whipple, 2012; Lague, 2014; Scherler et al., 2014). We note that





*A* and $k_{sn}$ do not correlate in the dataset (Kendall tau rank correlation coefficient = -0.036) so that the parameters can be
assumed to be independent. Due to the large number of data points, we binned the data into 150 logarithmically distributed
bins according to the ratio of *A* and $k_{sn}$. We calculated the mean and standard error of valley width and the ratio of *A* to $k_{sn}$ for
each bin. Our proxy parameter is therefore given by

$$\frac{q_L}{U} \sim m = \overline{\left(\frac{A}{k_{sn}}\right)}.$$

(20)

Here, the overbar denotes the mean. Before binning, we removed all data points with a steepness index smaller than 1 $m^{1.8}$.
This threshold steepness corresponds to a channel slope of 0.2% at a drainage area of 1 $km^2$, which we consider as unrealistic
for an active mountain belt.

## 4   Results
### 4.1   Model predictions

The model predicts that valley width evolves logarithmically between two limits (eq. 16, Fig. 4). For zero hillslope sediment
supply $q_H$, the model predicts an asymptotic approach to the unconfined channel-belt width $W_0$ for large values of the mobility-
uplift parameter $M_U$, which corresponds to large values of the lateral transport capacity $q_L$ or small values of uplift rate *U*
(eq. 16). When uplift rate is high or lateral transport capacity is low (small values of $M_U$), the equation levels off at the channel
width $W_C$. For intermediate $M_U$, valley width increases logarithmically as the lateral-transport capacity increases or uplift
decreases. For finite hillslope sediment supply $q_H$, the unconfined valley width reached at large $M_U$ is correspondingly reduced
(Fig. 4, dotted line). As $M_U$ increases, the effect of a lateral sediment supply in narrowing the valley increases. However, the
relative reduction of the excess width (*W*-$W_c$) by a lateral sediment supply relative to a case with $q_H = 0$ is constant, independent
of $M_U$.





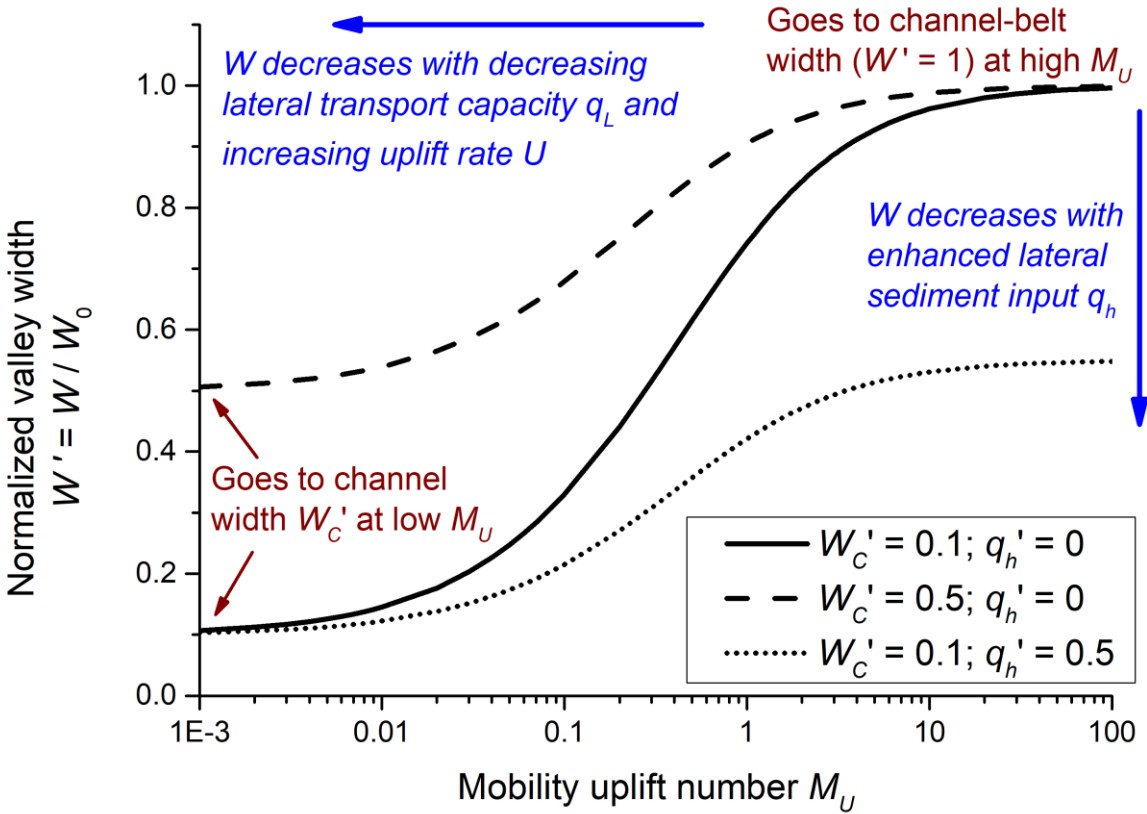

**Fig. 4: Evolution of dimensionless valley width as function of the mobility-uplift number $M_U$, predicted by eq. (16). An increase in $M_U$ corresponds either to an increase in lateral transport capacity, $q_L$, or a decrease in uplift rate $U$. A change in the relative channel width $W_c'$ affects the left-hand limit (solid, dashed and dash-dotted lines), while a change in the relative hillslope supply $q_h'$ affects the right-hand limit (solid and dotted lines).**

### 4.2 Comparison to data

Our valley width model can closely trace the relationship between valley width and $m$ in the experimental, the Tian Shan, and the Himalaya data sets (Fig. 5, Table 3). For the experimental data set we obtained an effective unconfined valley width $W_0$ = 2.7 m, and a channel width $W_C$ = 0.29 m, with an $R^2$ of 0.90 (Fig. 5A). The value of $W_0$ = 2.7 m corresponds to the total width of the basin available for bevelling (Fig. 3 in Bufe et al., 2016a) and is about 22% higher than the inferred actively bevelled width (median Af/L = 2.22 m). The channel width varies in the experiments and often there are multiple channels. The fitted value is thus an effective value. It is around half of the minimal observed channel width of 0.50-0.56 m when flow concentrates into a single channel.



For the Tian Shan data set, we fitted data from the north and south separately. For the north, we obtained an unconfined valley width $W_0$ = 495 m, and a channel width $W_C$ = 243 m, with an $R^2$ of 0.80 (Fig 5B). For the south, we obtained an unconfined valley width $W_0$ = 971 m, and a channel width $W_C$ = 22 m, with an $R^2$ of 0.97 (Fig 5B). Note that fitted unconfined valley and channel widths represent averages for streams with very different drainage areas (cf. Table 2).

The data from the Himalaya (Clubb et al., 2023b) shows considerable scatter, and we performed two fits to binned means of the datapoints, rather than to all of the data (Fig. 5C). The first fit includes all data, and yielded an unconfined valley width $W_0$ = 266 m, a channel width $W_C$ = 63 m, with an $R^2$ of 0.63. For the second fit, we excluded all bins with a mean valley width above 300 m. These high valley width appear as outliers at in the data (Fig. 5C), and we suggest that the widest valleys were dominantly formed or modified by processes other than the fluvial bevelling assumed in the model, for example, glacial erosion, alluvial valley infilling, or large-scale landsliding (e.g., Harbor, 1992; Montgomery, 2002; Stolle et al., 2017; Zakrzewska, 1971). For this fit, we obtained an unconfined valley width $W_0$ = 236 m, a channel width $W_C$ = 51 m, with an $R^2$ of 0.91 (Fig. 5C). The estimate of channel width is likely affected by the 30-m resolution of the digital elevation model that underlies the dataset, which hampers the identification of valley that are narrower than about 60 m.





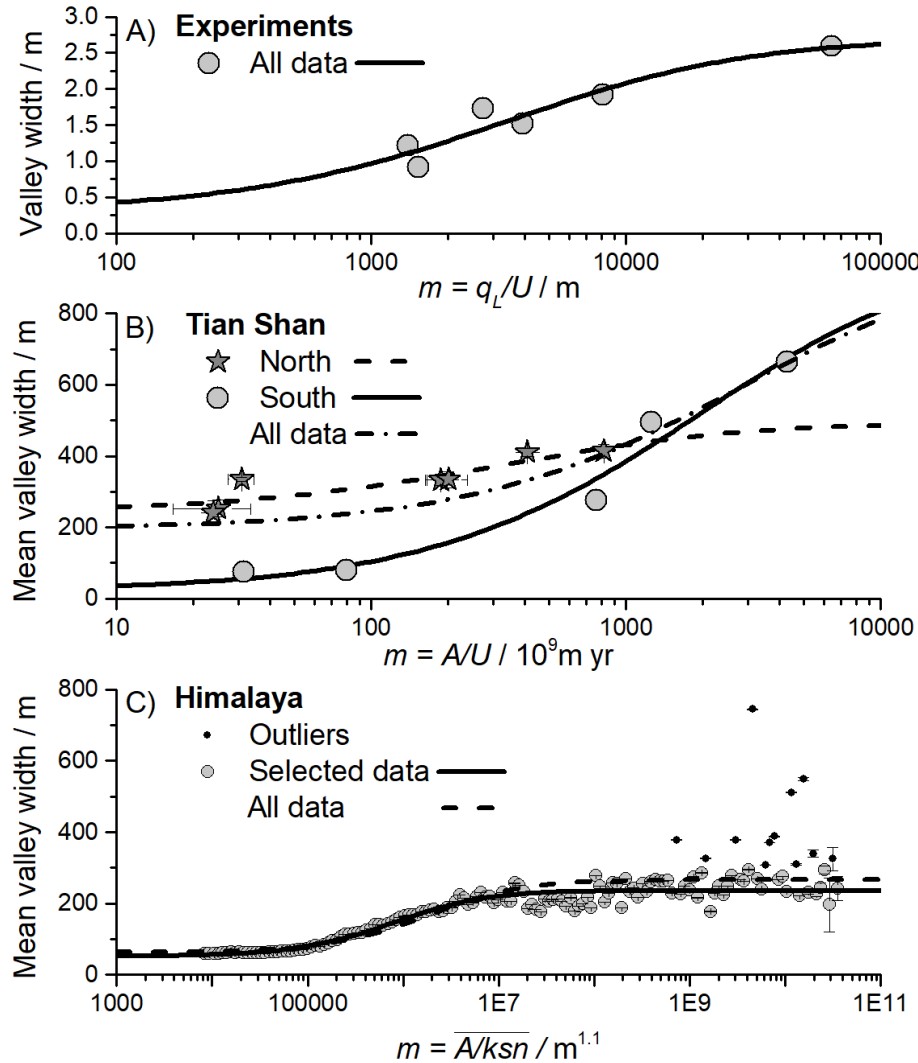

**Fig. 5: Valley width as a function of the proxy *m* for the ratio of lateral transport capacity and uplift. Lines give fits according to eq. (17); all fit parameters are listed in Table 3. (A) Data from the analogue experiments by Bufe et al. (2016a). Mean valley width is**
**calculated from the bevelled area of the fold divided by the length of the uplifted area of 0.5 m (Table 1). The proxy for the ratio of lateral transport capacity, $q_L$ and uplift rate *U* is given in eq. (18). (B) Valley width in the Tian Shan (Table 2) as a function of the ratio of drainage area, used as a proxy for $q_L$, and uplift rate (eq. 19). Values for the north (dark stars, dashed line) and south (grey dots, solid line) are treated separately. The dash-dotted line shows a fit to all data. (C) Mean valley width shown as a function of the ratio of drainage area *A* and the steepness index $k_{sn}$, with the latter assumed to linearly scale with uplift rate (eq. 20). Error bars**
**show the standard error of the mean for all values within a bin. We assumed that values of $k_{sn} < 1$ are unrealistic. If all remaining data points are included, the fit yields $R^2 = 0.63$ (dashed line). However, assuming that valleys with a mean width above 300 m are dominantly formed by processes other than fluvial bevelling, some of the data points can be treated as outliers (black circles). The remaining data points yield $R^2 = 0.91$ (grey circles, solid line). Note that the inferred channel widths $W_c$ are likely affected by 30-m resolution of the digital elevation model that underlies the dataset.**




**Table 3: Fit values for the data tests.**

| Dataset | Group | $\overline{W_0}$ / m | $\overline{W_C}$ / m | $a$ | $R^2$ |
|---|---|---|---|---|---|
| Experiments | All | 2.7 | 0.29 | $2.13 \times 10^{-4}$ | 0.90 |
| Tian Shan | North | 495 | 243 | $0.34 \times 10^{-9}$ yr$^{-1}$ | 0.80 |
| | South | 971 | 22 | $0.21 \times 10^{-9}$ yr$^{-1}$ | 0.97 |
| | All | 961 | 196 | $0.12 \times 10^{-9}$ yr$^{-1}$ | 0.70 |
| Himalaya | All | 266 | 63 | $4.66 \times 10^{-5}$ m$^{-0.1}$ | 0.63 |
| | Reduced | 236 | 51 | $9.85 \times 10^{-5}$ m$^{-0.1}$ | 0.91 |

### 4.3 Downstream variation of valley width

Our model was derived by considering valley formation in a cross section. However, the model can also yield predictions on how valley width develops with changing drainage area along a channel, because channel width, $W_C$, unconfined channel-belt width, $W_0$, and lateral transport capacity, $q_L$, all depend on water discharge. We can compare the predictions for the scaling between valley width and drainage area from our model with existing data. Based on empirical observations, multiple authors (e.g., Beeson et al., 2018; Brocard & van der Beek, 2006; Clubb et al., 2022; Langston & Temme, 2019; May et al., 2013;

Snyder et al., 2003; Tomkin et al., 2003) have suggested that valley width scales with drainage area according to a power law of the form

$$W = k_W A^\omega.$$

(21)

Here, we compiled information on the scaling exponent from various studies (Table 4) and compare them with predictions

from our model.

**Table 4: Scaling of valley width and drainage area**

| River or stratigraphic unit | Prefactor $k_W$ / km$^{(1-2\omega)}$ | Scaling exponent $\omega$ | $R^2$ | Reference |
|---|---|---|---|---|
| Sweden Creek | 1.25 | 0.37 | 0.36 | Beeson et al., 2018 |
| Rock Creek | 1.31 | 0.77 | 0.79 | Beeson et al., 2018 |
| Herb Creek | 1.04 | 0.62 | 0.39 | Beeson et al., 2018 |
| Scare Creek | 1.11 | 0.3 | 0.13 | Beeson et al., 2018 |
| Charlotte Creek | 0.97 | 0.62 | 0.51 | Beeson et al., 2018 |
| Halfway Creek | 0.94 | 0.6 | 0.41 | Beeson et al., 2018 |
| Dean Creek | 0.74 | 1.05 | 0.74 | Beeson et al., 2018 |
| Big Sand Creek | 0.97 | 0.93 | 0.69 | Beeson et al., 2018 |



| | | | | |
|---|---|---|---|---|
| c3 sandy limestone | 47 | 0.34 | 0.22 | Brocard & van der Beek, 2006 |
| n6 marls | 457 | 0.11 | 0.07 | Brocard & van der Beek, 2006 |
| n5 massive limestone | 41 | 0.18 | 0.11 | Brocard & van der Beek, 2006 |
| n4 well-bedded limestone | 47 | 0.21 | 0.14 | Brocard & van der Beek, 2006 |
| n3 marly limestone | 28 | 0.41 | 0.61 | Brocard & van der Beek, 2006 |
| n2 marls | 63 | 0.29 | 0.29 | Brocard & van der Beek, 2006 |
| n1 well-bedded limestone | 21 | 0.40 | 0.42 | Brocard & van der Beek, 2006 |
| j6 limestone | 41 | 0.18 | 0.08 | Brocard & van der Beek, 2006 |
| j5 limestone | 54 | 0.18 | 0.12 | Brocard & van der Beek, 2006 |
| j2-j4 black shales | 111 | 0.31 | 0.46 | Brocard & van der Beek, 2006 |
| Crane Creek | 2.09 | 0.24 | 0.71 | Clubb et al., 2022 |
| Bullskin Creek | 1.34 | 0.3 | 0.55 | Clubb et al., 2022 |
| Sugar Creek | 0.25 | 0.36 | 0.26 | Clubb et al., 2022 |
| Gilbert's Big Creek | 0.04 | 0.49 | 0.62 | Clubb et al., 2022 |
| Elisha Creek | 2.31 | 0.2 | 0.32 | Clubb et al., 2022 |
| Flat Creek | 0.01 | 0.56 | 0.78 | Clubb et al., 2022 |
| Hell for Certain Creek | 5.68 | 0.14 | 0.39 | Clubb et al., 2022 |
| Rockhouse Creek | 1.77 | 0.23 | 0.38 | Clubb et al., 2022 |
| Short Creek | 192.87 | -0.09 | 0.08 | Clubb et al., 2022 |
| Stinnett Creek | 305.51 | -0.13 | 0.20 | Clubb et al., 2022 |
| Cumberland River | 0.08 | 0.37 | 0.46 | Clubb et al., 2022 |
| Kentucky River | 0.14 | 0.33 | 0.37 | Clubb et al., 2022 |
| Licking River | 2.71 | 0.22 | 0.19 | Clubb et al., 2022 |
| Guyandotte River | 0.69 | 0.26 | 0.34 | Clubb et al., 2022 |
| Little Kanawha River | 0.16 | 0.34 | 0.51 | Clubb et al., 2022 |
| 1 (undisturbed) | 0.027 | 0.41 | 0.64 | Harel et al., 2022 |
| 2 (undisturbed) | 0.18 | 0.54 | 0.93 | Harel et al., 2022 |
| 3 (undisturbed) | 0.19 | 0.54 | 0.94 | Harel et al., 2022 |
| 4 (undisturbed) | $2.43\times10^{-3}$ | 0.26 | 0.45 | Harel et al., 2022 |
| 5 (beheaded) | $3.33\times10^{-3}$ | 0.23 | 0.42 | Harel et al., 2022 |
| 6 (beheaded) | $1.33\times10^{-3}$ | 0.15 | 0.37 | Harel et al., 2022 |
| 7 (beheaded) | $1.48\times10^{-3}$ | 0.18 | 0.73 | Harel et al., 2022 |
| 8 (reversed) | $4.76\times10^{-9}$ | -0.74 | 0.37 | Harel et al., 2022 |
| 9 (reversed) | $1.38\times10^{-9}$ | -1.00 | 0.23 | Harel et al., 2022 |
| 10 (reversed) | $3.67\times10^{-6}$ | -0.24 | 0.69 | Harel et al., 2022 |
| 11 (reversed) | $3.58\times10^{-6}$ | -0.18 | 0.26 | Harel et al., 2022 |



| 12 (reversed) | $1.05 \times 10^{-8}$ | -0.56 | 0.64 | Harel et al., 2022 |
|---|---|---|---|---|
| Elk Creek | | 0.6 | | May et al., 2013 |
| Harvey Creek | | 0.6 | | May et al., 2013 |
| Sedimentary | 61.1 | 0.34 | 0.45 | Schanz & Montgomery, 2016 |
| Basalt | 28.4 | 0.22 | 0.40 | Schanz & Montgomery, 2016 |
| Oat | 0.028 | 0.41 | 0.51 | Snyder et al., 2003 |
| Kinsey | 0.0072 | 0.50 | 0.62 | Snyder et al., 2003 |
| Shipman | 0.0066 | 0.50 | 0.15 | Snyder et al., 2003 |
| Gitchell | 0.118 | 0.32 | 0.34 | Snyder et al., 2003 |
| Horse Mtn. | 0.026 | 0.42 | 0.50 | Snyder et al., 2003 |
| Hardy | 0.181 | 0.29 | 0.36 | Snyder et al., 2003 |
| Juan | 0.012 | 0.46 | 0.69 | Snyder et al., 2003 |
| Clearwater River | 2.8 | 0.76 | | Tomkin et al., 2003 |

In the limits of the model for small and large $M_U$, we expect that valley width approaches respectively the channel width $W_C$
or the unconfined valley width $W_0$. Therefore, at these limits, the scaling between valley width and drainage area should follow
the scaling between drainage area and respectively $W_C$ and $W_0$. Because the latter parameter scales with flow depth (eq. 8), we
need to consider the drainage area scaling for channel width, and flow depth. Channel width $W_c$ and flow depth $h$ also
commonly scale with drainage area (e.g., Ferguson, 1986; Gleason, 2015; Leopold & Maddock, 1953; Park, 1977; Rhodes,
1978). The $W_c$-$A$ scaling exponent typically varies between about 0.3 and 0.6, with a most commonly cited value of 0.5 (e.g.,
Ferguson, 1986; Gleason, 2015; Leopold & Maddock, 1953). In turn, the $h$-$A$ scaling exponent typically varies between 0.2
and 0.5, with a most-commonly cited value of 0.4 (e.g., Ferguson, 1986; Gleason, 2015; Leopold & Maddock, 1953). However,
for both exponents, values that are higher or lower than the stated range are not uncommon. For example, Park (1977) gives a
range between 0.09 and 0.70 for the $h$-$A$ scaling exponent, and 0.03 and 0.89 for the $W_c$-$A$ scaling exponent from a global data
compilation. Rhodes (1978) gives a similar range between 0.01 and 0.84 for the $h$-$A$ scaling exponent, and 0 and 0.84 for the
$W_c$-$A$ scaling exponent. As a result, based on eqs. (8) and (15), our model predicts that valley width $W$ should increase with
drainage area $A$ according to a power law with an exponent between 0.03 and 0.9, and the most likely value of 0.4 - 0.5 (Park,
1977; Rhodes, 1978). The range of the $W$-$A$ scaling exponents $\omega$ compiled from the literature (Table 4) corresponds well to
these expected ranges (Fig. 6).

The scaling factor $k_0$ between channel-belt width and flow depth (eq. 8) cannot be accurately constrained with the presently
available data. For the experimental dataset, Bufe et al. (2016a) estimated the flow depth at 7.5 mm, which implies $k_0 = 321$
(Table 3). A value of $k_0$ of the order a few hundred seems also to be reasonable when considering the field data (cf. Table 3).





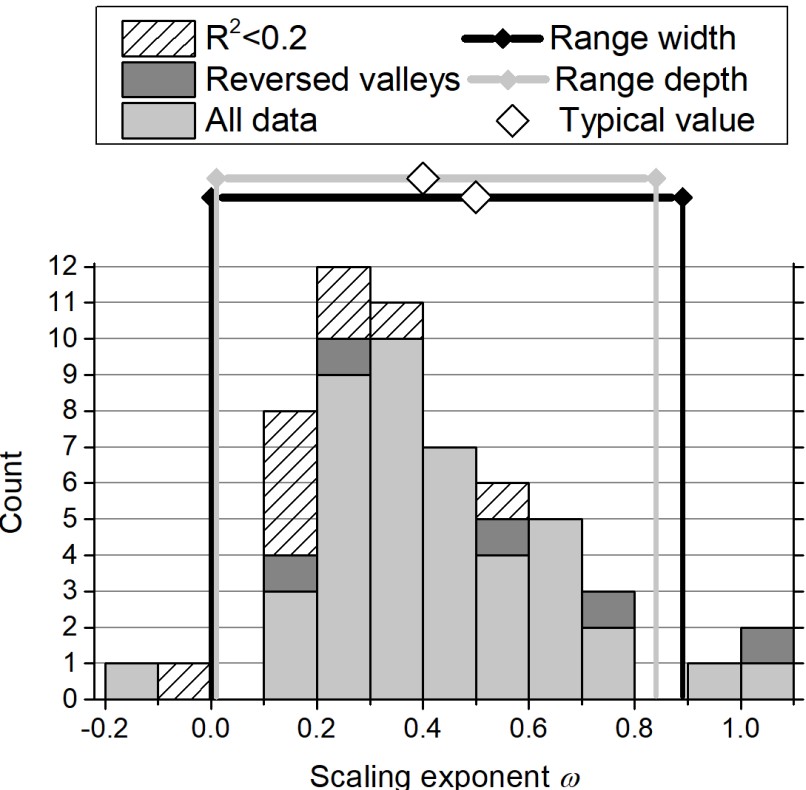

**Fig. 6: Histogram of 57 valley width to drainage area scaling exponents $\omega$ (eq. 21), compiled from the literature (Table 4). Light grey blocks give all data from regular valleys with a correlation coefficient $R^2$ between valley width and drainage area exceeding 0.2, striped blocks give data with $R^2 < 0.2$, and dark grey blocks correspond to reversed valleys reported by Harel et al. (2022), where tilting reversed the flow direction of the river. The range of scaling exponent values for flow depth (grey; 0.01-0.84) and channel width (black; 0-0.89) reported by Park (1977) and Rhodes (1978) are also indicated (cf. eq. 8).**

## 5    Discussion

### 5.1    Model concept

Our model predicts the scaling of valley or channel-belt width with flow depth, bank height and channel width (eq. 8), and how this valley width is modulated by uplift and lateral hillslope sediment supply (eq. 15). It is, essentially, a deterministic steady-state model for valley width, building on the stochastic concept of a river channel migrating across an alluviated floodplain and switching direction according to a Poisson process. The model reproduces the relationships between valley width and the ratio of lateral channel mobility and uplift in three separate data sets (Fig. 5). In addition, it predicts a range of scaling exponents between valley width and drainage area that is consistent with those observed in natural settings (Fig. 6). As such, it has quantitative explanatory power that expands previous efforts focusing on the transient widening phase (Martin et al., 2011, Hancock & Anderson, 2002), on non-uplifting valleys (Tofelde et al., 2022), or empirical (e.g., Langston & Temme, 2019; Brocard & van der Beek, 2006; Beeson et al., 2019), numerical (Langston & Tucker, 2018), and qualitative (Clubb et





al., 2023a) descriptions of valley formation. In addition, the model, in principle, encompasses all currently known controls on
steady state valley width (cf. Martin et al., 2011; Tofelde et al., 2022), and yields a wealth of testable predictions. For example,
it yields an equation for channel-belt width (eq. 8), and predicts that valley width is controlled by four dimensionless numbers
(eq. 15). Our model predicts that fluvial-valley width is controlled by both climatic and tectonic conditions, but is explicitly
independent of lithology at steady state. While tectonics come into the model via the uplift rate, climate appears indirectly,
either as a control on unconfined channel-belt width in the limit of low uplift rate, or as a control on channel width in the high
uplift rate limit. Likewise, lithology exerts an implicit, indirect control by changing channel width (see Section 5.3).

The good fit of our model to multiple datasets ranging from rivers crossing a single fold to an entire orogen suggest that many
valleys, especially at small drainage areas and / or high uplift rates, are formed to first order by laterally migrating rivers. We
note that the Himalayan data are characterized by large scatter that can arise from multiple factors, such as variations in
sediment grain size and lithology, unequal distributions of rainfall, non-steady state valleys, response to transient, non-uniform
uplift, or the dominance of other processes than fluvial bevelling in setting valley width. Yet, our model provides an excellent
fit to the binned means of the data ($R^2 = 0.91$), especially when bins with mean valley width exceeding about 300 m are
excluded (Fig. 5C). Hence, we suggest that the model can be applied to a wide variety of physiographic settings. The two field
data sets that we used for our tests originate from active mountain belts, the Tian Shan (Section 3.2) and the Himalaya (Section
3.3). As such, these channels are likely controlled by bedrock and probably cannot be considered as fully alluvial rivers. We
suggest that in bedrock rivers, valley widening occurs during times when there is no active bedrock incision, and the bedrock
floor of the valley is covered by sediment, such that the river behaves like an alluvial river (cf. Turowski et al., 2013). This
means that the fill needs a depth equal to or exceeding the flow depth. As the river sweeps laterally through the sediment fill,
it occasionally erodes the walls and removes the sediment that is provided from the walls by hillslope erosion (Tofelde et al.,
2022) until a steady-state valley width is achieved. As such, the composition and erodibility of the valley walls should affect
the speed of widening during the transient phase, but not the steady state valley width.

### 5.2 Valley width scaling with drainage area

Our model predicts that valley width is equal to channel-belt width $W_0$ in the limit of low uplift rates, and to channel width $W_C$
in the limit of high uplift rates (eq. 15). These two limits are important to consider, because they potentially apply to a large
proportion of data in natural settings. In all datasets, as well as in the model, the logarithmic dependence of valley width on
lateral transport capacity and uplift rate exists across 2-3 orders of magnitude of the $q_L/U$ ratio ($m$) (Fig. 5). In natural settings,
this ratio can span up to eight orders of magnitude (Fig. 5C), so that most natural valleys sit at either the $W_c$ or $W_0$ limit. Using
these two limits, we will in the following discuss the scaling relationship between valley width and drainage area.

Channel-belt width is proportional to the square of flow depth $h$ divided by bank height $H_0$ (eq. 8). In a situation without uplift,
it seems reasonable to assume that bank height, on average, corresponds to flow depth. After all, the river cannot deposit
sediment at heights above its flow surface, and if the bank were lower, the channel overtops, widens, and becomes shallower
or finds a different course. Yet, there may be some variability in the bank heights due to autogenic changes between incision





and deposition phases in and along the river channel (Mizutani, 1998; Bufe et al., 2019). Such changes lead to variations in channel width and depth, which can lead to variations in bank height encountered by the river throughout the floodplain, in

turn affecting the *W-A* area scaling exponent. Further, along-stream variations in uplift or lateral channel mobility $q_L$ – which is affected, for example, by grain size – may affect the scaling exponent. Overall, the range of observed valleys of the scaling exponent (Table 3) matches the expected range from hydraulic geometry quite well (Fig. 6). High values may arise from specific local conditions, other active processes, or along-stream gradients in the control variables such as uplift rate. These would need to be investigated locally in specific case studies.

In model construction, we have not explicitly considered the response of channel geometry to uplift. It is widely accepted that incising channels are narrower and deeper than non-incising or depositing channels at the same water discharge and sediment supply (e.g., Lavé & Avouac, 2001; Turowski, 2018; Yanites et al., 2010). Channel-belt width $W_0$ dominantly depends on flow depth (eq. 8), and valley width is close to channel-belt width for low uplift rates. When the mobility-uplift number $M_U$ is large, an increase in uplift rate may thus indirectly lead to an increase in valley width, because the river responds to the increase in

uplift with a decrease in flow width and an increase in flow depth. We expect that this counter-intuitive result is applicable only in rare circumstances, when a change in uplift rate is large enough to cause an observable change in valley width due to the change in flow depth, but not so large such that the direct control of uplift rate on valley width dominates.

### 5.3 Lithological controls on steady state valley width

Multiple observations point to a lithological control on valley width and indicate that valleys carved into more erodible rock tend to be wider than valleys carved into less erodible lithologies (e.g., Bursztyn et al., 2015; Keen-Zebert et al., 2017; Moore, 1926; Schanz & Montgomery, 2016). Brocard & van der Beek (2006) and Langston & Temme (2019) observed higher scaling exponents in the relationship of valley width and drainage area in softer rocks compared to harder rocks. These observations contrast with the absence of a correlation of valley width with lithological units in the Himalaya, as reported by Clubb et al.

(2023a). Our findings from the Himalaya suggest that a majority of valleys may be close to one of the valley-width limits, where valley width approaches either the channel width $W_C$ or the channel belt width $W_0$ (Fig. 5C). It is likely that lithology influences the former limit, because as the width of bedrock channels in mountain regions increases with increasing erodibility (e.g., Turowski, 2018). Further, our model suggests that the channel width – and therefore any lithologic control on channel width – affects the shape of the model curve beyond the limit of small mobility-uplift numbers (cf. the solid and dashed lines

in Fig. 4). From field observations, we expect that channel width varies by a factor below ten for various lithologies (e.g., Ehlen and Wohl, 2002; Spotila et al., 2015). For example, Montgomery & Gran (2001) reported a halving of channel width of a river crossing from a limestone into a granite reach, and Spotila et al. (2015) observed a maximum factor of five for different lithologies for channel width after normalizing for drainage area. As such, the observed lithological dependence of valley width (e.g., Brocard & van der Beek, 2006; Langston & Temme, 2019; Schanz & Montgomery, 2016) is consistent with our

model. In addition to the channel width, lithology may also influence the balance between hillslope sediment supply and removal. In summary, we posit that the scaling relationship between valley width and drainage area is implicitly dependent on





lithology in our model, via the dependence of channel width on lithology. This dependence can be expected to emerge when scaling relationships in individual valleys are studied (as done by Brocard & van der Beek, 2006, and Langston & Temme, 2019), but should disappear when data from many different valleys are averaged within a regional perspective (as done by Clubb et al., 2023a).

### 5.4    Comparison to previous models

Our model concept both contrasts with and builds on previous models of fluvial valley formation (Fig. 7; cf. Clubb et al., 2023a; Hancock & Anderson, 2001; Martin et al., 2011; Tofelde et al., 2022). We classify existing models using two criteria (Fig. 7). First, we distinguish transient from steady state models. Second, we distinguish models that emphasize vertical from those that emphasize lateral processes. This latter distinction essentially corresponds to the alluvial and bedrock categories proposed by Clubb et al. (2023a), in which the alluvial model emphasizes vertical processes and the bedrock model lateral processes.

In the *alluvial model* (Fig. 7A), valley width is set by depositing sediment into a pre-existing V-shaped valley, created during an earlier incision phase. Because the channel bed is located on the surface of the sediment fill, valley width is set passively to the width determined by the slanting valley walls at the height of the sediment fill. Valley width is thus set by the angle of repose and the amount of sediment delivered from upstream. The alluvial model includes both transient and steady state elements.

The *eternal widening* (EW) model (Fig. 7B) is a transient model emphasizing lateral processes. It assumes that the valley floor grows by fluvial undercutting of the valley walls and subsequent wall collapse (e.g., Hancock & Anderson, 2002; Malatesta et al., 2017; Martin et al., 2011; Langston & Tucker, 2018). It exists in several variants that differ in the precise formulation of the erosion model and the description of channel dynamics. In the EW model, valley width is a function of the widening or wall-erosion rate integrated across the duration of widening (Hancock and Anderson, 2002; Suzuki, 1982). Although the widening rate decreases as valleys grow wider through time – because the fraction of time the river spends cutting into the walls declines (Hancock & Anderson, 2002) – the valley never reaches a steady state width. Valley width thus depends on the widening rate and the time since the last incision event. Tofelde et al. (2022) added the notion of channel-independent hillslope sediment delivery to the EW model (Fig. 7C). In the case of this *lateral-flux steady state model*, valleys can achieve a steady state width when sediment supply from hillslopes and evacuation by the river are balanced (cf. eq. 12).

Within the present contribution, we add the concept that rivers randomly change the direction of migration, according to a Poisson process (Fig. 7D). This yields an average switching timescale for channel migration, setting an average bevelled width. Assuming that steady state width corresponds to the mean behaviour of the stochastic model, channel belts reach a steady state width even without confinement. This steady state channel-belt width gives a maximum width for fluvial valleys, which can be reduced due to uplift or lateral hillslope sediment input (Fig. 4). This model can be termed a *deterministic version of the Poisson model*.



A fully *stochastic Poisson model* has not been treated within the present contribution, but is implicit in the assumptions underlying the derivation (Fig. 7E). Due to the random motion during the Poisson process, the channel can venture beyond the steady state width predicted by the deterministic Poisson model. Essentially, once the steady state width has been reached, the channel may push beyond the valley boundaries on either side of the valley. This increases the width on one side, but leads to less frequent visits of the flood plain on the other side. Thereby, these parts of the valleys are abandoned. We expect that this

effect results in a slow lateral drift of the areas frequently revisited by the river after the steady state channel belt has been established. In an uplifting area, the valley floor would thus shift laterally over time, without changing its width. In an area without uplift, the effect leads to never-ending and ever-slowing increase of valley width over time. It thus reconnects to the eternal widening model, providing a similar outcome with a different mechanism.

Finally, we note that valley width could be set or modified by processes other than lateral erosion of the valley walls by the

river, or deposition and evacuation of sediment. These could be, for example, back-weathering of the walls (e.g., Krautblatter & Moore, 2014; Moore et al., 2009; Tofelde et al., 2022), downstream-sweep erosion of the river controlled by upstream conditions (Cook et al., 2014), large-scale landsliding (e.g., Beeson et al., 2018; Stolle et al., 2017), or glacial processes (e.g., Montgomery, 2002; Zakrzewska, 1971). These processes likely contribute to the scatter observed in the data and may explain some of the observed outliers (Fig. 5C).






**Figure 7: Overview of the existing models, organised into groups according on whether they emphasize vertical or lateral processes (vertical axis), and whether they yield a transient or steady state valley width (horizontal axis). See text for model descriptions.**



It is beyond the scope of this paper to comprehensively test the models against each other and against data. Instead, we want to briefly outline the boundary conditions that are necessary for the different models to apply. Clubb et al. (2023a) found a strong inverse correlation of valley width with the channel steepness index $k_{sn}$ in their data (Fig. 4C), but no correlation with lithological units. They argued that this observation rules out the dominance of lateral processes (Fig. 7B), and instead suggested that the alluvial model (Fig. 7A) prevails, where valley width is mainly set by sediment deposition and evacuation

in a previously existing V-shaped valley. Clubb et al. (2023a) suggested that high uplift rates lead to elevated channel bed slopes, and that the river responds by deposition to build these slopes in a pre-existing valley. However, substantial deposition of sediment following the incision of a valley can only occur (i) if there is a substantial increase in the ratio of upstream sediment supply to water discharge, (ii) if the relative base level rises, or (iii) if the stream is disconnected from the original base level, which is possible for example when the channel is blocked by a massive landslide (e.g., Korup, 2006). Thus, we

expect that, in an uplifting landscape, filled valleys generally present transient features. Case (i) can occur if either climatic conditions change (affecting both sediment supply and discharge), or if upstream uplift rates increase (affecting downstream sediment supply). A comparison of different valleys is then only meaningful if a similar change occurs in all basins at the same time. This seems unlikely given the wide range of climatic and tectonic conditions within the Himalaya. Case (ii) can occur if the base level uplift rate increases or if the uplift rate throughout the catchment decreases. Because an entire region is

considered, at least some of the catchments would necessarily see opposite effects. Further, a comparison of different catchments would only be meaningful if the change in uplift occurs at the same time. In case (iii), the river disconnects from the downstream base level, and as a result, the channel is insensitive to the uplift. Assuming the upstream regions keep eroding at the same rate as prior to the disconnection, the amount of deposited sediment – and therefore valley width – should scale positively with uplift rather than negatively, contradicting the observation from the data from the Himalaya.

In contrast, our model implies that the role of uplift is to increase the thickness of alluvium that the river has to move through when migrating laterally – thereby slowing the lateral back-and-forth movement of the river and narrowing the valley. As argued in Section 5.3, our model is consistent both with an absence of lithological control on the steady state valley width in a regional perspective (Clubb et al., 2023a), and emerging lithological controls in the scaling relationships of individual valleys (Brocard & van der Beek, 2006; Langston & Tucker, 2019). As explained above, we propose that the erosion rate of valley

walls by the river modulates the transient rate of widening up to the steady state. In turn, at steady state the river does not actively erode valley walls, but the steady state valley width is limited either by the rate of lateral sediment input from hillslope erosion processes (Tofelde et al., 2022) or the likelihood of channel switching within the valley. Of course, in an uplifting setting, the river has to incise bedrock. In our concept, incisional and widening phases of the river are separated, as has been suggested previously (e.g., Martin et al., 2013; Hancock & Anderson, 2001), but it is not necessary that the incisional phases

at all times carve deep V-shaped valleys that are subsequently filled up. Many mountain rivers feature a closed sediment cover during low and intermediate flows (e.g., Tinkler and Wohl, 1998; Turowski et al., 2013), with a thickness of a few meters – enough for a river to sweep back and forth across the valley within the alluvium. In turn, vertical incision dominantly occurs during large floods (e.g., Cook et al., 2018; Lamb and Fonstad, 2008). Turowski et al. (2013) suggested that rivers alternate





between the deposition and evacuation of sediment during floods and intermediate flow, because transport capacity and
sediment supply both depend on, but scale differently with discharge. Whether a particular river is 'flood-cleaning', i.e., it evacuates sediment during big events, or 'flood-depositing', i.e., it deposits sediment during big events, depends on site-specific conditions relating to hydrology, substrate, climate and channel morphology.

In cases where valleys are deeply infilled (Blöthe et al., 2014; Wang et al., 2014), the river moves laterally through the fill and widens the valley when encountering the bedrock walls. If the valley width imposed by the fill – as in the alluvial model
(Fig. 7A) – exceeds the steady state width, infilled valleys may be wider than predicted by equation (15). This can potentially explain some of the observed outliers (Fig. 5C).

## 6 Conclusion

Within this contribution, we have derived a physics-based steady-state model for the width of channel belts and river valleys. In agreement with previous suggestions, we assume that valleys widen by river undercutting of valley walls. We add the notion
of random changes in the river's direction of motion, which can be described by a Poisson process. We link the probability of switches per unit to the river's lateral mobility (Bufe et al., 2019), and channel depth and width. We derive a deterministic steady state model for fluvial valley width that can account of all currently known controls, including channel lateral mobility (Bufe et al., 2019), discharge and sediment supply (e.g., Beeson et al., 2018; Tomkin et al., 2003), lateral hillslope sediment supply (Tofelde et al., 2022), uplift or incision rate (Bufe et al., 2016a; Clubb et al., 2023a), the absence of a correlation with
lithology in a regional perspective (Clubb et al., 2023a), and lithological controls on scaling relationships of individual valleys (Brocard & van der Beek, 2006; Langston & Tucker, 2019). The model predicts that for low uplift rates, valley width is equal to channel-belt width, and for high uplift rates, it is equal to channel width. A logarithmic function connects these two limits for intermediate uplift rates (Fig. 4, eq. 15). The model corresponds well to field and experimental data of valleys in uplifting settings (Fig. 5).

The model yields a wealth of quantitative predictions that can, in principle, be tested against experimental and field data. Its analytical equation can be used to track valley width in models of river corridors (e.g., Heimann et al., 2015; Wickert & Schildgen, 2019) or entire landscapes (e.g., Barnhart et al., 2020; Gailleton et al., 2023). It thus may allow for more comprehensive descriptions of mountain landscapes or the interaction of rivers with their floodplains.

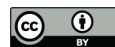



**Symbols & Notation**

| | |
|---|---|
| $\lambda$ | Rate parameter of Poisson processes describing the switch in the direction of river motion [T$^{-1}$] |
| $a$ | Scaling parameter (varying units) |
| $A$ | Drainage area [L$^2$] |
| $A_f$ | Area actively reworked by channel prior to uplift in experiments [L$^2$] |
| $c$ | dimensionless constant of order 1 [-] |
| $h$ | Flow depth [L] |
| $H_+$ | Height of the river bank in the direction of river motion [L] |
| $H_-$ | Height of the river bank opposite the direction of river motion [L] |
| $H_0$ | Constant bank height in conditions without tectonic uplift [L] |
| $k$ | Dimensionless constant [-] |
| $k_0$ | Dimensionless constant, defined by $c/k$ [-] |
| $k_{sn}$ | Normalized steepness index [m$^{0.9}$] |
| $k_W$ | Pre-factor in the power law scaling between valley width and drainage area [L$^{1-\omega}$] |
| $m$ | Proxy that scales with $q_L/U$ (varying units) |
| $M_U$ | Mobility-uplift number, $M_U = q_L/UW_0$ [-] |
| $q_H$ | Rate of lateral sediment supply from hillslopes or valley walls per channel length [L$^2$ T$^{-1}$] |
| $q_H'$ | Normalized hillslope sediment supply, $q_H' = q_H/q_L$ [-] |
| $q_L$ | Lateral-transport capacity, i.e. the amount of sediment that the channel can move by lateral erosion per unit channel length per unit time [L$^2$ T$^{-1}$] |
| $P$ | Fraction of time that a river spends at any of its channel walls or valley margins [-] |
| $\Delta t$ | The characteristic length of time the river moves on average in the same direction [T] |
| $T_f$ | Timescale over which $A_f$ was reworked [T] |
| $U$ | Uplift rate [L T$^{-1}$] |
| $v$ | Lateral speed of the river as it reaches valley-floor margins, i.e. wall toes [L T$^{-1}$] |
| $V$ | Lateral migration speed, i.e. the speed of river migrating back and forth across the valley floor [L T$^{-1}$] |
| $W$ | Valley-floor width [L] |
| $W'$ | Normalized valley width, $W' = W/W_0$ [-] |



| $W_c$ | Width of the river channel [L] |
|---|---|
| $W_c'$ | Normalized channel width, $W_C' = W_C/W_0$ [-] |
| $\overline{W_C}$ | Average channel width [L] |
| $W_0$ | Channel-belt width or unconfined valley width [L] |
| $\overline{W_0}$ | Average channel-belt width [L] |
| $\omega$ | Scaling exponent in the power law scaling between valley width and drainage area [-] |

**Data availability**

Raw data for the experimental datasets are stored on the SEAD repository of Bufe et al. (2016b) with the identifier http://dx.doi.org/10.5967/M0CF9N3H. Derived quantities have been compiled from Bufe et al. (2016a,b) and Bufe et al.

(2019). All data necessary for reproducing the results are also given in Table 1. The mapped channel widths and auxiliary data from the Tian Shan are given in Table 2. The valley width data from the Himalaya extracted by Clubb et al. (2023a) and auxiliary data can be found on the repository of Clubb et al. (2023b) with the identifier https://doi.org/10.15128/r2z890rt27d .

**Competing interests**

At least one of the (co-)authors is a member of the editorial board of Earth Surface Dynamics.

**Author contributions**

All authors have contributed to the conceptualization of the study, model development, data analysis, and writing.

**Acknowledgments**

Fiona Clubb and her co-authors graciously shared their data from the Himalaya prior to publication. We thank Fergus McNab for discussions and feedback on the manuscript. Sophie Katzung mapped valley width in the northern Tian Shan.

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
