# Peer review of "A process-based model for fluvial valley width"

_EGUsphere, 2023_

## Referee Comment (RC1)

**Review of Turowski et al. (2023) A process-based model for fluvial valley width**

The authors seek to understand how valley width develops under different climate, tectonic, and lithologic conditions by formulating a deterministic equation for valley width. By developing an equation, the authors allow individual components of the system to be analyzed. As an example, the authors show how analysis of their valley width equation reconciles research showing that lithology controls valley width and research demonstrating a lack of lithologic control.

This manuscript is well reasoned and is in a near publishable state. The work is a significant contribution to geomorphology and, as the authors point out, could have implications for ecology, archaeology, and fish biology. I have a few comments that represent minor edits.

In line 145, the authors state that channels change direction when there is too much sediment from the opposite bank to transport. Please provide observations or citations to support this. While it makes intuitive sense, this assumption underlies the derivation for $\lambda$, and so I'd like to see more formal support for it.

The confined and unconfined valley width formulations assume that valley widening occurs separately from incision; the authors further suggest in line 465 that "valley widening occurs during times when there is no active bedrock incision, and the bedrock floor of the valley is covered by sediment" (also stated in line 63). However, actively meandering bedrock channels show evidence for both lateral widening and vertical incision during the same period (e.g., Limaye and Lamb, 2016; Merritts et al., 1994). In this case, lithology affects vertical incision rates, and is not just a factor in the erodibility of valley walls. I think (if I am reasoning this correctly) that this does not affect your ultimate statement in line 470 that lithology affects the speed of widening but not the steady state width. However, I suggest revising the statements in line 465 and line 63 to acknowledge that bedrock rivers can simultaneously incise and migrate laterally.

In unconfined valley settings and wide confined valleys, meander migration and cutoffs result in rapid switches in channel position as well as upstream migrating knickpoints (e.g., Finnegan and Dietrich, 2011). Those stochastic knickpoints may affect valley width by first causing an increase in transport capacity, followed by an increase in bank height and increased lateral sediment supply once the knickpoint has passed. Could this process have a significant effect on the analytic valley width solution?

Line 331: Citation needed to say that erosion rates are a good proxy for uplift in the Himalayas

Last, the authors describe the model as process-based (title, line 226), deterministic (line 441), and physics-based (line 613), and should choose a consistent terminology. I do not agree that this is a process-based model, as it does not include processes of valley widening beyond broad consideration of the effects of sediment supply vs transport (i.e., a process-based model might be more focused on meander process). Particularly since many of the variables in the three

study sites are estimated empirically, I suggest describing this as an 'empirical and deterministic' model.

References:

Finnegan, N. J. and Dietrich, W. E.: Episodic bedrock strath terrace formation due to meander migration and cutoff, Geology, 39, 143–146, https://doi.org/10.1130/G31716.1, 2011.

Limaye, A. B. S. and Lamb, M. P.: Numerical model predictions of autogenic fluvial terraces and comparison to climate change expectations, Journal of Geophysical Research: Earth Surface, 121, https://doi.org/10.1002/2014JF003392, 2016.

Merritts, D. J., Vincent, K. R., and Wohl, E. E.: Long river profiles, tectonism, and eustasy: A guide to interpreting fluvial terraces, Journal of Geophysical Research: Solid Earth, 99, 14031–14050, https://doi.org/10.1029/94JB00857, 1994.

---

## Author Comment (AC1)

We thank the reviewers for the constructive comments. Below, the comments are given in normal fond, while our replies are in *italics*.

Review #1 Sarah Schanz

Review of Turowski et al. (2023) A process-based model for fluvial valley width
The authors seek to understand how valley width develops under different climate, tectonic, and lithologic conditions by formulating a deterministic equation for valley width. By developing an equation, the authors allow individual components of the system to be analyzed. As an example, the authors show how analysis of their valley width equation reconciles research showing that lithology controls valley width and research demonstrating a lack of lithologic control.
This manuscript is well reasoned and is in a near publishable state. The work is a significant contribution to geomorphology and, as the authors point out, could have implications for ecology, archaeology, and fish biology. I have a few comments that represent minor edits.
*Thank you for the constructive comments.*

In line 145, the authors state that channels change direction when there is too much sediment from the opposite bank to transport. Please provide observations or citations to support this. While it makes intuitive sense, this assumption underlies the derivation for $\lambda$, and so I'd like to see more formal support for it.
*To our knowledge, there is no prior work on this issue and the argument is original to our work. As such, it provides a testable prediction within our model framework. We are not aware of work directly pertaining to this point (hence, our label as 'postulate'). However, it is a common observation that lateral sediment input (e.g., by landslides or tributaries) pushes rivers to the side, and we have added a statement on this and some references.*
*The text now reads: "We did not find documented observations of this notion, and a thorough investigation will need to be done in the future. Yet, it is commonly observed that lateral sediment input pushes rivers towards the opposite bank (e.g., Cruden et al., 1997; McClain et al., 2020; Savi et al., 2020)."*

The confined and unconfined valley width formulations assume that valley widening occurs separately from incision; the authors further suggest in line 465 that "valley widening occurs during times when there is no active bedrock incision, and the bedrock floor of the valley is covered by sediment" (also stated in line 63). However, actively meandering bedrock channels show evidence for both lateral widening and vertical incision during the same period (e.g., Limaye and Lamb, 2016; Merritts et al., 1994). In this case, lithology affects vertical incision rates, and is not just a factor in the erodibility of valley walls. I think (if I am reasoning this correctly) that this does not affect your ultimate statement in line 470 that lithology affects the speed of widening but not the steady state width. However, I suggest revising the statements in line 465 and line 63 to acknowledge that bedrock rivers can simultaneously incise and migrate laterally.
*We agree that bedrock rivers can simultaneously incise vertically and erode bedrock walls laterally, especially when the river is meandering actively. However, would this lead to valley widening? In this particular case, the valley floor would be sloped (the so-called slip-off slope), and the lateral river migration would not result in a flat valley floor. This seems a little bit a matter of definition, potentially*

*invoking a threshold of the lateral slope of the bedrock valley floor. In any case, we believe that the mechanics in this case work slightly differently than outlined in our current model.*

*We note that we did not state that vertical incision and lateral erosion never occur simultaneously, but that "valley widening occurs during times when there is no active bedrock incision, and the bedrock floor of the valley is covered by sediment". We have now added the word 'dominantly', and added a few references. In particular, the in press paper by Langston & Robertson, ESPL 2023, is in line with our expectation, as they showed in experiments that high sediment supply and laterally mobile channels are necessary to create wide bedrock valleys.*

*The passage now reads:* "We suggest that in bedrock rivers, valley widening occurs dominantly during times when there is no active bedrock incision, and the bedrock floor of the valley is covered by sediment, such that the river behaves like an alluvial river (cf. Shepherd, 1972; Turowski et al., 2013). This notion is in line with recent experiments of Langston & Robertson (2023), who found that high sediment supply, sediment cover on the bed, and laterally mobile channels in the alluvium are needed for the formation of wide bedrock valleys."

In unconfined valley settings and wide confined valleys, meander migration and cutoffs result in rapid switches in channel position as well as upstream migrating knickpoints (e.g., Finnegan and Dietrich, 2011). Those stochastic knickpoints may affect valley width by first causing an increase in transport capacity, followed by an increase in bank height and increased lateral sediment supply once the knickpoint has passed. Could this process have a significant effect on the analytic valley width solution?
*Very interesting question! We would say: it depends on the averaging timescale. For very long time scales, it should average out. For intermediate time scales, it will have a local effect. Given that we base our model on the consideration of a cross-section and a steady state achieved over long time scales, we have not added a discussion to the manuscript.*

Line 331: Citation needed to say that erosion rates are a good proxy for uplift in the Himalayas
*We added three references for this statement, Scherler et al., 2014, Hodges et al., 2004, and Lenard et al., 2020.*

Last, the authors describe the model as process-based (title, line 226), deterministic (line 441), and physics-based (line 613), and should choose a consistent terminology. I do not agree that this is a process-based model, as it does not include processes of valley widening beyond broad consideration of the effects of sediment supply vs transport (i.e., a process-based model might be more focused on meander process). Particularly since many of the variables in the three
study sites are estimated empirically, I suggest describing this as an 'empirical and deterministic' model.
*We disagree somewhat with this statement. Our model is based on considerations of first principles and generic physics-based arguments, and the arguments we use are entirely independent of the data we test it against. Further, our model, as depicted in equations 15 and 16 only contains parameters that have a direct physical interpretation. All parameters could at least in principle be measured in the field and in experiments (however, we are very aware of the many practical issues in doing this, for example, the variability of flow depth h in natural settings). There are no lumped parameters that need empirical calibration, and only two dimensionless scaling parameters (c in equation 2 and k in equation 7). Both of these can be, in principle, measured in the lab or field. The need for the fit parameter m (eq. 17) arises from the incompleteness of the currently available data sets, and is not inherent in the model set up. As such, we do not think that the model is 'empirical' in any conventional sense of the word.*

*One can argue about what is a process. In geomorphology, we often have a scale dependence of what is seen as 'process'. Case in point, meandering, used as an example by the reviewer above, depends on the process of grain entrainment by turbulent flows. As such, meandering could be viewed as an emergent behavior, not a process.*
*In response to this comment, we have changed 'process-based' to 'physics-based'.*
*We did not remove the use of the term 'deterministic' – in line 441, this term is identified as a label, and the thinking behind this label is explained in detail in section 5.4.*

Review #2 Sebastién Carretier

This manuscript presents a new model to explain the functional relationships between the width of a valley W and several parameters such as drainage area and tectonic uplift. It is very well written, with very broad implications for understanding landscape dynamics over geological time. The widening of valleys is still poorly understood, and such a model provides a framework that could enable this element of the landscape to be interpreted quantitatively in terms of climate and tectonics. The fit between the data and the model is remarkable, even if some of this fit is due to the adjustment of certain parameters. This suggests that the scaling relationships between W and the various ingredients of the model are correct. The model is based on a number of simplifying assumptions, starting with the assumption that valley widening reaches a limit through time. This model assumes and applies to a stationary state of W. To derive the model, J. Turowski and colleagues follow an original approach, starting with a simple definition of W and gradually integrating the ingredients that lead to the final equation linking W with the other parameters. However, I did not always fully understand the derivation of these equations and I have some doubts about certain assumptions. The rest of my report is more a discussion of these misunderstandings than a challenge to the model. These comments can be used to improve the presentation of the model.
*Thanks for the supportive comments.*

I'm not sure I fully understand Equation (1) linking valley width W with lateral migration velocity V. If I understand correctly, this equation assumes that the width W is a constant width, obtained after a time Deltat. If I still understand correctly, this time is defined as the average reoccupation time of the same site by the channel, in particular the edge of the valley.
*Equation 1 is a general formulation that does not necessitate a steady state valley width. It states that valleys result from lateral motion of a river channel, i.e., valley is wherever the river has been at some point. This is in line with many prior formulations. Speed could be variable in time. The timescale could be infinite, if there is no change in direction. Equation 1 would still be the same. The steady state width then comes with the assumption that the rivers switch directions at regular intervals Deltat (see answer to next comment).*
*We added: "Equation (1) is a general formulation for valley formation by fluvial bevelling, which allows, for example, for variable V."*

However, a little above, it is stated that the width is set by the average time during which the channel migrates in the same direction. How are these two times related? What is the underlying vision: a channel that migrates for a certain time in one direction and then abruptly changes position (through

avulsion or some other process)? In other words, why is the maximum bound in the integral of Equation (3) a mean time and not an infinite time, since we are looking for a stationary solution?

*Delta_t is the average time that the channel migrates in one direction. Essentially in the steady state model, we make the 'effective' assumption that the river moves for a constant time in one and then for the same time in the other direction. And so on. As such, once the channel has traversed the entire valley once, there is no further widening, because it changes direction exactly at the time when it touches one of the channel walls. We then identify the relevant timescale with the average switching time of the stochastic process (equation 2), acknowledging that they scale and may not be exactly the same (hence the dimensionless scaling factor c).*

*The upper bound is not infinite because the channel switches direction. And during the times it moves within the valley, it does not widen the valley. An infinite upper bound would imply that the channel never switches direction. Note that in a fully stochastic treatment of channel switching, there is probably some drift in that actively-reworked steady-state width. Such a drift is observed in experiments (e.g. Fig. S4 in Bufe et al., 2019), and we are working on how to include it in the modelling of channel switching. This drift is briefly mentioned in the discussion of the fully stochastic model in section 5.4.*

*We revised the statement after introducing eq. 2 to: "We proceed by considering the average behaviour of the channel belt, essentially making the assumption that the channel switches direction of migration at regular intervals $\Delta t$."*

Why is an integral needed here rather than writing directly that the width is determined by the product of the speed of migration in one direction and the average time Deltat during which the channel is in contact with the edge of the valley (=V.Deltat) ?

*Yes, in the simple case that V is independent of time (i.e., a constant), the width is given by V\*Delta_t. This, essentially, is the result for the unconfined case given in eq. 4. Yet, the integral formulation is more general than stating V\*Delta_t. We use it directly when incorporating uplift in equation 10. Further, the integral in eq. 1 opens the possibility for future extensions of the model, for example, for integrating over variable discharge.*

*We added: "Equation (1) is a general formulation for valley formation by fluvial bevelling, which allows, for example, for variable V. For constant V, $W = V\Delta t + W_C$."*

Does it make a difference if the total water discharge splits between multiple channels? Croissant et al (2018 JGRES) show that continuity of transport capacity as the valley widens requires a change from a single channel to multiple channels. Does the number of channels in the valley influence the likelihood of widening the valley?

*This is a very interesting question! We cannot fully answer this at the moment, but we can make a few generic and observational statements.*

*First, we do not fully understand the controls on $q_L$, what we call the lateral transport capacity. We do know that it depends on, for example, water discharge, sediment supply and grain size, but the functional relationships are debated and have not been finalized (see Bufe et al., ESPL 2019 for an investigation of experimental data, and a wider discussion). Because we do not fully understand $q_L$, we cannot make any reliable statements on how it changes when the same amount of water and sediment is split up into different channels.*

*Second, multiple channels add a considerable complexity. For example, there is no requirement that all channels at all times migrate into the same direction, implying that the channels interact and their number, size, etc. evolves over time. Incorporating this into the model would require a number of*

*additional assumption (on channel merging, splitting, migration...), and a scheme of keeping track of their motion within the cross section to map their individual contributions to valley widening. This is beyond our first simplified attempt to address the problem, but yields interesting questions for future research.*
*Third, Bufe et al.'s (2016) experiments, which we compared to the model (section 3.1), frequently did feature multiple channels. Still, the model provides a reasonable explanation of the data, potentially indicating that incorporating multiple channels would yield similar scaling relationships to our simple model.*
*We have added a paragraph to section 5.1.*

Are there arguments to justify a Poissonian distribution of waiting times? Would another type of distribution (from one valley to another) dramatically change the model's predictions?
*Note that the waiting times in a Poisson process are exponentially distributed. The number of events per time is Poisson distributed. This corresponds to a uniform distribution of events within time.*
*A Poisson process arises if: 1) events are independent and identically distributed (this implies that there is no history dependence for the particular events, i.e., the channel switches direction independently of when it switched last time or how often it has switches before), 2) two events cannot occur at the same time (i.e., there is always a finite but arbitrarily small waiting time between two events), and 3) the average rate of events is a constant. These are the formal assumptions that we make and state.*
*For the steady state model, the underlying distribution is not relevant, as long as it has a well-defined mean over the time scales that are relevant for valley formation. It does make a difference for the transient development of the valley towards the steady state, and for the valley drift in a full stochastic treatment. We are currently preparing a follow-up paper focusing on transient evolution.*
*We added: "As such, the likelihood of the switches in direction have no history dependence. The stated conditions mean that the switching of directions is described by a Poisson process with rate parameter $\lambda$ [T$^{-1}$] that quantifies the mean number of switch events per unit time."*
*Please also refer to the replies above.*

In Equation (5) the characteristic shift time is proportional to the ratio between the lateral transport capacity and the cross-sectional volume of a channel. I would have rather written that it is proportional to the ratio between the longitudinal flux Qs of sediment passing through the channel and the volume of the channel (e.g. Sun et al., 2002, WRR). This flux can be related to the upstream uplift rate but also the drainage area and could therefore change the scaling relationship between width W, the steepness index and the drainage area. Similarly, avulsion frequency increases with sediment flux Qs (e.g. Bryant et al., 1995, Geology). If Lambda depended on Qs, would this dramatically change the model's predictions?
*$q_L$ depends on upstream sediment supply (see Bufe et al., 2019). So, the dependence on Qs is implicit in this equation, and it does not change the central results. Note that the implicit dependence may be weak (see Bufe et al., 2019).*
*We note here that the controls on $q_L$ are not fully resolved (see Bufe et al., 2019 for a wider discussion).*

In equation (15), two different definitions of the probability of a channel "touching" the edge of the valley seem to be combined. The first corresponds to a Poissonian process (Equations (2)-(11)) and the second to a homogeneous probability which only depends on the ratio between the width of the channel and the width of the valley. Is there not a conflict between these two visions and a contradiction in combining them in equation (15)?

*The Poisson process gives a rate for the number of switches. In contrast, P is the fraction of the time that river spends at a valley wall while moving in the direction of this wall. This would be the probability of observing valley widening when looking at the river at a random time, while the river is not yet at a steady state. The two notions do not describe the same thing. Note that we did not use the term 'probability' when defining P.*

*In fact, P is necessarily zero in steady state in the 'deterministic' model approach (the river does not spend any time widening the valley anymore). The function for P would look different in a fully stochastic treatment.*

*In equations 12 to 15, we use P to incorporate lateral sediment supply. The interpretation of the term is the similar as before, and we have slightly redefined it after eq. 12, now stating: "P [-] is the fraction of time that the river spends at the valley walls with a direction of motion towards them".*

Would it be possible to test the validity of the model by analysing the sudden changes in valley width when a river moves from a confined to an unconfined situation as it leaves a mountain range?

*Yes, this may provide suitable natural experiments. One needs to be a bit careful when considering depositional systems, especially where avulsion is an important process (e.g., at the mountain fronts, rivers often construct alluvial fans). Net depositional systems have not yet been incorporated into our model framework, and while it should be possible to describe valley formation due to avulsing rivers by a similar framework as used here, the meaning of q_L would be different (compare to the discussion of Bufe et al., 2019).*

*We have added in section 2.1: "We assume that depositional systems do not naturally lead to incised valleys. We will thus consider graded or incising channels, and assume that they move laterally by bank erosion, rather than avulsion."*

The model predicts functional relationships between valley width and several geomorphological parameters, but not lateral erosion rates.

*We have not delved into this particular issue in the present paper, because we are concerned with the steady state solution, in which lateral erosion rates are necessarily zero. It is pretty straight forward to derive some results for the case with non-zero lateral erosion rates, and we have provided some discussion of this for the model predecessor (see Tofelde et al., AGU Advances, 2022). We are currently preparing a follow-up paper focusing on transient and non-steady evolution of valleys within our model framework.*

Some data from Zavala et al (GRL 2021) seem consistent with this model. For example, Zavala et al. found a very low valley-side erosion rate for the valley located upstream of a knick-zone in the Tana valley in Chile. The valley is not very wide and is not deeply incised into the pampas (this point is an outlier in the erosion rate versus W graphs). This seems consistent with the fact that the valley has reached a state of equilibrium between qL and qH which determines Wc, and for which lateral erosion is necessarily low.

*Thanks for pointing this out. We are aware of the Zavala et al. paper. In fact, in our previous paper (Tofelde et al., AGU Advances, 2022), where we first suggested the balance between qL and qH, we provided a discussion of and a comparison with the Zavala data. The bulk of that comparison can be found in the supplement S1 of that paper. For your reference, we include here the relevant figure and caption (see Tofelde et al., 2022, supplementary material S1 for more details).*

[Figure]

**Figure S2**. Testing our model against data from the Chiza Canyon in the Atacama Desert. (a) Valley width, valley-flank height, total erosion rates of valley flanks based on $^{10}$Be (numbers in m/My), and valley-flank gradient have been measured at six locations within the Chiza Canyon, Chile (Zavala et al., 2021). In response to Miocene surface uplift on the forearc a knickzone has been migrating upstream and is currently located close to sample site 6. Zavala et al. (2021) connect the higher valley-flank erosion rates at location 5 and 6 to the ongoing response of the river to knickzone migration. (b) Conceptual relationship of how the two components of vertical river incision, $E_V$, and horizontal hillslope erosion, $E_H$, contribute to the total hillslope-erosion rate measured by $^{10}$Be, $E_T$ (after Zavala et al., 2021). From $E_T$ and $S_H$, the parameter $E_H$ can be calculated as long as $E_V$ is small or can be independently estimated. (c) Comparison between measured valley width by Zavala et al. (2021) and predicted valley width based on our model. Note that we estimated several required parameters based on proxy data (see text for details). Hence, predicted valley widths have no uncertainties, have no metric unit and do not necessarily scale 1:1 with measured valley width.

*(Figure and caption from Tofelde et al., AGU Advances, 2022, https://doi.org/10.1029/2021AV000641, supplementary material S1)*

Specific comments

Line 59, "rock type" and weathering.
*Added.*

Lines 71-72 These two sentences about the climate seem to me to contradict each other.
*Changed to: "which have frequently been suggested to form in response to cyclic climate change"*

Line 215. What is Wu (not used afterwards)? Is it Wc?
*This was a left-over from a previous version with slightly different notation. Changed to: "can be identified with the width of an uplifting valley".*

Equations (19) and (20). Is the dependence of Wc on A taken into account in the model? And if not, wouldn't this change the model's predictions about the scaling relationship between W and A?
*It depends on what one is interested in. For the scaling of valley width with drainage area along an individual stream, the dependence of W on A would definitely have to be taken into account. We do provide a brief discussion of this point in section 4.3. When dealing with a regional picture, not considering upstream-downstream connectivity of valleys, using channel width provides a simpler approach, mainly because channel width depends on drainage area (and other parameters) in a complicated way that is not fully understood. A beautiful feature of the model equation (eq. 15 and 16) is that the in the two limits, there is only a dependence on channel and unconfined valley width. This means that fits to data converge to the true mean of the data set in these limits.*

Line 557. I'm not sure I understand. Do you mean that the width increases so slowly in the EW model that this amounts to fixing the width of the valley?
*The point here is that in a fully stochastic treatment, for a case without uplift, the valley keeps widening infinitely, as has been suggested in the EW model. The reason for eternal widening, however, differs between the two models.*
*Within the EW model, the widening rate depends on the formulation of the fraction of time that the channel spends incising the valley walls. In previous work, this was given as the ratio of channel to valley width, implying that the width increases with the squareroot of time. Using our formulation of the probability (eq. 13), the EW model predicts a logarithmic increase of valley width in time. The stochastically driven increase in valley width in our model also increases with the squareroot of time, as for random walks (derivations will be published in an upcoming paper).*
*In any of these formulations, it depends on the observation time and the time since the start of the development of the valley, whether a valley is increases its width so slowly that it is perceived to not increase at all.*
*We slightly revised the sentences to read: "In an uplifting area, the valley floor would thus shift laterally over time, without changing its width. Valleys in an area without uplift will widen indefinitely at an ever-slowing pace. This is analogous to the prediction of the eternal widening model, yielding a similar outcome with a different mechanism."*

Line 591: "thereby slowing the lateral back-and-forth movement of the river and narrowing the valley ":
On the contrary, it could be argued that increasing Qs favours lateral mobility (Bryant et al., 1995) and widening (Baynes et al., 2020, ESPL).

*Here we comment on a slightly different issue. The sentence refers to the local effect of uplift in raising the material that the river has to erode through to move. Therefore, for a constant lateral transport capacity, increasing channel-bank heights will slow down the lateral velocity. It is possible that regional uplift impacts the downstream sediment discharge and modulates mobility – that could be modelled by modulating the value of q_L. See also previous comments on the controls on q_L (Bufe et al., 2019).*
*We added:* "This uplift effect does not consider changes in downstream sediment discharge that could be driven by increased landscape-scale erosion rates. Such an effect can be modelled in the form of modulating lateral transport capacity $q_L$."

Line 592: "absence of lithological control on the steady state valley width in a regional perspective". I am not sure that the model calibration exercise in figure 5C really demonstrates this. Insofar as the fitted data is an average per mobility number value class, there can only be one value of Wc for the whole data set. Shouldn't you separate the data into different datasets by lithology category to check that W does not depend on the lithology globally?
*This statement references the discussion in Section 5.3.* ("As argued in Section 5.3, our model is consistent both with an absence of lithological control [...]"). In that section, w*e argued that lithology likely controls width of individual valleys through its control on Wc. However, when fitting a large regional dataset, such as the fit to the data by Clubb et al. (2023), we fit an effective average of all channel widths that, therefore, averages across all lithologic differences. It is of course possible to look into the lithologic control in the data by Clubb et al. (2023), for example by separating the data by lithologic category as you suggest. Such analysis is beyond the scope of the paper.*
*We modified the sentence to read:* "As argued in Section 5.3, our model is consistent both with an absence of lithological control on the regionally averaged steady state valley width (Clubb et al., 2023a), and with emerging lithological controls in the scaling relationships of individual valleys (Brocard & van der Beek, 2006; Langston & Tucker, 2019)."

I wish you good luck for the revision.
*Thank you for the constructive comments*

Sébastien Carretier

---

## Author Response (AR2)

Thanks, Simon, for the detailed comments. Our replies are given in *italics*.

General remarks
1. *We added another study by Lifton et al. (2009) that we previously overlooked and that investigates local lithological controls on valley width.*
2. *We added 'hydrology and flood dynamics' as an additional relevant topic at the end of the first paragraph, slightly revised the statements and bolstered up the references.*
3. *Regarding the meandering literature mentioned at several points by the AE – we were aware of many of the mentioned papers (Constantine et al. 2014 was actually cited in the introduction). We were not aware of the paper of Ielpi and Lapôtre 2019, so thanks for this suggestion. The crucial points for us are that (1) the controls on migration speed are debated and none of the mentioned papers provides a full picture. A problem here is that many of the potential drivers are cross-correlated. For example, Ielpi and Lapôtre use channel width to normalize, which is also correlated, for example, with slope and water discharge. As such, a purely empirical approach is not ideal for clearly discriminating control variables. We provided a comprehensive discussion of potential controls of migration speeds in the Bufe et al. 2019 paper, including several approaches for dimensional analysis, and a discussion of the experimental and field data available at the time of publication. (2) Within in the context of the valley width model, the relevant parameter is lateral transport capacity qL, not lateral migration speed, and the precise controls on qL / speed are not of first order relevance for the outcomes of model (as long as we agree that speed V and qL are related by qL = V\*H+; see Bufe et al. 2019 for evidence and discussion).*
   *We have now added a couple sentences after the introduction of eq. 3:*
   *"Both V and $q_L$ are determined by hydraulic and sedimentological boundary conditions. The precise controls have not yet been completely resolved (e.g., Bufe et al., 2019; Constantine et al., 2014; Ielpi & Lapôtre, 2019; Wickert et al., 2013), but are not directly relevant for the remainder of the derivation."*

16 Strange syntax. I don't think you need the "once".
*Deleted as suggested.*

43 keep this if you want, but I don't think it is needed.
*Deleted 'grounds' as suggested.*

102 "forth and back" sounds very strange to a native English speaker. If you don't wish such persons to be confused I suggest "back and forth".
*Changed as suggested (although, 'forth and back' seems more logical, because one needs to go forth before being able to come back…).*

104 The basic premise is that the upper value of valley width is determined by the channel belt width, when the channel is unconfined. The model then goes on to discuss lateral velocity of the river in such a setting.  I find it very strange that there is no reference at this point to the river meandering literature. Surely the information about lateral migration rates from that literature is relevant here?
*See general comment #3.*

119 This is a strange way of saying something that seems trivial. Maybe I have misunderstood. You are just saying there are waiting times longer than the mean waiting time? Is this not obvious? I suggest altering the wording here so it is clear what you are trying to say.

*Maybe we are making things a bit more complicated than necessary. The point is that in a fully stochastic version of the model, switching time is obviously not constant, and the steady state valley width sets to some 'effective' or 'characteristic' switching time. This characteristic switching time should scale with the mean switching time, but cannot be expected to be exactly equal to it. The parameter c is meant to account for this. The sentence was revised to:*

"As such, the valley sets its width to some effective lateral migration time that can be expected to scale with the mean waiting time, but is not necessarily exactly equal to it."

126 Support (or not) for this assumption can be found in the meander literature. It would be useful to say how dependent the results are on this assumption.
*See general comment #3.*

131 I am a bit confused by this image. The "unconfined" image shows a braided river. This images does delineate the floodplain as part of the channel belt. The confined river shows a weakly braided river with a floodplain being included. I don't think these two images are comparing the same thing.
I think panel b would be more useful if you showed a relative elevation raster derived from lidar, which makes it clear that the river migrates back and forth along the floodplain (many examples are shown on social media, if you are into that sort of thing).
*This comment sparked quite some discussion amongst us about the definitions of channel belt, floodbelt, floodplain and so on. We have resolved to not go into that in the present paper, and have instead revised the figure to focus on the terms that are actually relevant for the model.*
*The new version of the figure is:*

[Figure]

141 Again, some mention of the meander literature would be useful here. Do you not think it is relevant? If not, why not? There have been several studies on meander migration rates that link meander velocity to sediment fluxes (not lateral...from upstream) e.g. constantine et al (2014) Nature geosciences
*See general comment #3.*

166 This paper: https://www.nature.com/articles/s41561-019-0491-7 says that h is a power law function of width, and velocity is also a power law function of width.
I don't expect a new model but some reference to this literature would be useful.
*Thanks, we had not seen this paper.*
*The geometrical relationship between channel width, flow depth, flow velocity, and discharge is known as hydraulic geometry and has been widely investigated since the 1950ies (the relationships were first described by Leopold and Maddock, cited elsewhere in our manuscript). The relationship between width and depth is related to this concept. The mean behavior suggests that the width-to-depth ratio is weakly dependent on discharge, with an exponent of ~0.1 when moving downstream and ~0.14 at a given cross-section. Please note that the power law exponents (and correspondingly the aspect ratio) vary widely between river systems (see our discussion on this issue in section 4.3 and 5.2, and the compilations of Park 1977 and Rhodes 1978 cited there). Further, this is a mean behavior, which can locally change due to instabilities, bank erosion and so on. Even in the paper suggested by the AE (Ielpi and Lapôtre, 2019), flow depth varies about an order of magnitude for a given width (see Fig. 3c). Further, the paper is not concerned with the channel switching direction of migration, for which we develop an argument at this particular point.*
*See also general comment #3.*

185 As long as the forcing is constant.
*Yes, that is correct. It is clearly stated in line 183, two sentences prior, in the opening sentence for the paragraph and section. The model framework would also allow to deal with non-uniform incision, but we do not explore this in the present paper. No changes.*

203 I think it would be helpful to the reader to spell this out: "as can be expected based on straight, entrenched rivers in rapidly uplifting landscapes." (or choose some statement as to why this is expected).
*Changed to "as can be expected for entrenched rivers in rapidly uplifting landscapes".*

209 I don't expect any changes here, but in the meander literature the sediment supply controls the meander rate, so it would be interesting to consider if higher sediment fluxes from upstream would modulate the rate. Do you think it plays a role?
*Yes, sediment supply is one of the controls on lateral transport capacity $q_L$. The precise controls on $q_L$ do not affect the outcomes in the way we have been using the model in the present paper. See also general comment #3.*

213 I would write out "dimensionless" here.
*Changed to "Here, P [-] is a dimensionless parameter denoting the fraction of time…"*

251 I think the fitting procedure should have more detail here.
$W_0$ and $W_C$ are dependent on drainage area, but it seems like from the results below lump valleys with different areas together. Can you please explain how/why this is done and justify it. And give a precursor to the later section specifically dealing with this issue.
*A justification is given in the subsequent sentences, lines 252-255 in the old manuscript, lines 260-263 (final manuscript) and lines 262-265 (tracked changes) in the revised version. Please let us know if this is not sufficient.*

252 delete 'fit'
*Deleted.*

341 It isn't clear here if the references specifically said that you could linearly fit erosion rate and k_sn in the himalaya, or if these are general statements. It should say so.
*Revised to:*
"Even though relationships between $k_{sn}$ and erosion rate are commonly fit with non-linear power laws, the scatter in most data sets make a linear fit equally appropriate, both in general (e.g., Kirby & Whipple, 2012; Lague, 2014) and for the Himalaya specifically (e.g., Lague, 2014; Scherler et al., 2014)."

346 Why is the exponent here different to the stated units of k_sn on the previous page?
*Typo, corrected now.*

354 I think this would be easier to understand if the second sentence started with: "These two limits are defined by..."
In its current state it takes some time to parse (for me at least).
*We have revised the sentences to:*
"At large values of the mobility-uplift parameter $M_U$, corresponding to large values of the lateral transport capacity $q_L$ or small values of uplift rate $U$, the model predicts an asymptotic approach to the unconfined channel-belt width $W_0$ for zero hillslope sediment supply $q_H$ (eq. 16). Conversely, when uplift rate is high or lateral transport capacity is low (small values of $M_U$), the equation levels off at the channel width $W_C$."

378 This seems quite wide. How does it compare to satellite imagery?
*A direct comparison is hard to make, because (i) likely flood width is relevant, and (ii) the fit value is an average of channel widths of channels at different drainage areas. Yet, the value seems reasonable. For example, the flood width at Manas (basin with the largest drainage area of 5541 km² from the northern group) can be estimated to up to 700 m, with about 350 m seeming a reasonable estimate within the valley.*
*Coordinates of all locations are given in Table 2.*

392 What an amazing result.
*It is, isn't it?*

435 Nice result.
*Thanks!*

508 The meandering literature has data on this and should be cited here.
*The statement here is within the context of our model. We added:* "In our model" *to the sentence. See also general comment #3.*
*Generally, there are a lot of as yet untested predictions and assumptions in the model. Suitable data to directly test them are rare, and test construction is not so straight-forward for field settings in particular (for example, which flow depths is relevant in a channel with varying discharge?). For the particular statement made, the proportionality of channel belt width and flow depth, we are planning to look at the experimental data by Limaye (2020) in a future publication.*

613 Not sure if I agree with this: the rate of tectonic convergence will be correlated across the range. Or is this not what you are arguing?

*This is a valid point. Yet, in tectonically active settings, uplift is often not continuous, but occurs along faults, and there may be opposing effects on basins draining, for example, to the hanging wall and the footwall. There is also typically a variation of tectonic activity across the strike of major structures. The key point we want to make is that it is unreasonable to expect uniform forcing over a region as large as the Himalayas. We have revised to give a more careful statement:*

"Case (ii) can occur if the relative base level uplift rate.  Further studies are necessary to investigate whether the spatial or temporal variations in uplift rate for catchments in the Himalaya are consistent enough to cause major base-level changes and sediment aggradation of V-shaped valleys that are correlated across the range."

---

## Author Response (AR3)

We thank the editors for their consideration. We have implemented two small editorial changes in the revised manuscript.

Line 305: The reference Bufe et al. (2016) was changed to Bufe et al. (2016a) to avoid ambiguity.

Line 910: we removed the hyperlink of the doi in the reference to Repasch et al.